# A non-mosaic transchromosomic mouse model of Down syndrome carrying the long arm of human chromosome 21

Yasuhiro Kazuki[1,2†]*, Feng J Gao[3†], Yicong Li[3], Anna J Moyer[3,4], Benjamin Devenney[3], Kei Hiramatsu[1], Sachiko Miyagawa-Tomita[5,6], Satoshi Abe[2], Kanako Kazuki[2], Naoyo Kajitani[2], Narumi Uno[1], Shoko Takehara[2], Masato Takiguchi[1], Miho Yamakawa[2], Atsushi Hasegawa[7], Ritsuko Shimizu[7], Satoko Matsukura[8], Naohiro Noda[8], Narumi Ogonuki[9], Kimiko Inoue[9], Shogo Matoba[9], Atsuo Ogura[9], Liliana D Florea[4], Alena Savonenko[10], Meifang Xiao[11], Dan Wu[12], Denise AS Batista[13], Junhua Yang[3], Zhaozhu Qiu[3], Nandini Singh[14], Joan T Richtsmeier[15], Takashi Takeuchi[16], Mitsuo Oshimura[1], Roger H Reeves[3,4]*

[1]Department of Biomedical Science, Institute of Regenerative Medicine and Biofunction, Graduate School of Medical Science, Tottori University, Yonago, Japan; [2]Chromosome Engineering Research Center (CERC), Tottori University, Yonago, Japan; [3]Department of Physiology, Johns Hopkins University School of Medicine, Baltimore, United States; [4]Department of Genetic Medicine, John Hopkins University School of Medicine, Baltimore, United States; [5]Department of Animal Nursing Science, Yamazaki University of Animal Health Technology, Hachioji, Tokyo, Japan; [6]Department of Physiological Chemistry and Metabolism, Graduate School of Medicine, The University of Tokyo, Tokyo, Japan; [7]Department of Molecular Hematology, Tohoku University Graduate School of Medicine, Sendai, Japan; [8]Biomedical Research Institute, National Institute of Advanced Industrial Science and Technology (AIST), Tsukuba, Japan; [9]Bioresource Engineering Division, RIKEN BioResource Research Center (BRC), Tsukuba, Japan; [10]Departments of Pathology and Neurology, John Hopkins University School of Medicine, Baltimore, United States; [11]Department of Neuroscience, John Hopkins University School of Medicine, Baltimore, United States; [12]Department of Biomedical Engineering, Zhejiang University, Hangzhou, China; [13]Department of Pathology, John Hopkins University School of Medicine, Baltimore, United States; [14]Department of Anthropology, Penn State University, State College, United States; [15]Division of Biosignaling, School of Life Sciences, Faculty of Medicine, Tottori University, Yonago, Japan; [16]Department of Anthropology, California State University, Sacramento, United States

*For correspondence:
kazuki@tottori-u.ac.jp (YK);
rreeves@jhmi.edu (RHR)

[†]These authors contributed
equally to this work

Competing interests: The
authors declare that no
competing interests exist.

Reviewing editor: Susan L
Ackerman, Howard Hughes
Medical Institute, University of
California, San Diego, United
States

**Abstract** Animal models of Down syndrome (DS), trisomic for human chromosome 21 (HSA21) genes or orthologs, provide insights into better understanding and treatment options. The only existing transchromosomic (Tc) mouse DS model, Tc1, carries a HSA21 with over 50 protein coding genes (PCGs) disrupted. Tc1 is mosaic, compromising interpretation of results. Here, we "clone" the 34 MB long arm of HSA21 (HSA21q) as a mouse artificial chromosome (MAC). Through multiple steps of microcell-mediated chromosome transfer, we created a new Tc DS mouse model, Tc (HSA21q;MAC)1Yakaz ("TcMAC21"). TcMAC21 is not mosaic and contains 93% of HSA21q PCGs that are expressed and regulatable. TcMAC21 recapitulates many DS phenotypes including anomalies in heart, craniofacial skeleton and brain, molecular/cellular pathologies, and impairments

in learning, memory and synaptic plasticity. TcMAC21 is the most complete genetic mouse model of DS extant and has potential for supporting a wide range of basic and preclinical research.

## Introduction

Human aneuploidy, a gain or loss of chromosomes, is associated with both birth defects and cancer (*Beach et al., 2017*). Most aneuploidies result in miscarriage. Down syndrome (DS) is caused by trisomy 21 and is the most common survivable aneuploidy, occurring in about 1 in every 800 in live births (*Driscoll and Gross, 2009*). Besides common facial and other physical features, people with DS have intellectual disabilities, a high risk of congenital heart disease and leukemia, and early onset dementia (*Korenberg et al., 1994*). A population-based study of 6300 infants with DS born from 1993 to 2003 in US shows that 95% had a complete freely segregating third copy of HSA21, 3% had partial trisomy 21 due to duplication of part of the chromosome, and 2% had mosaic DS in which some cells have three copies of HSA21 while others have two copies (*Shin et al., 2010*). Both translocation DS and mosaic DS are rare and typically have milder phenotypes compared with those who inherit a complete HSA21 (*Prasher, 1995*; *Chandra et al., 2010*).

Trisomic mouse models have made a major contribution to DS research. Davisson's Ts65Dn mouse was the first viable DS model (*Davisson et al., 1990*; *Davisson et al., 1993*), and the demonstration that features directly comparable to those in DS occurred in mice with trisomy for a number of genes orthologous to HSA21 changed the paradigm for research in this area (*Reeves et al., 1995*). Ts65Dn has been the most broadly used model for DS research for more than two decades, up to and including crucial support for clinical trials (*Braudeau et al., 2011*). However, advances in genomics have defined HSA21 and its mouse orthologs much more precisely, and a better genetic representation is necessary to support current research efforts (*Hattori et al., 2000*; *Reinholdt et al., 2011*; *Duchon et al., 2011*; *Gardiner et al., 2003*).

There are a number of technical arguments for a new DS model to meet current challenges, but perhaps the most compelling global issue relates to pre-clinical drug testing. Testing of potential treatments in Ts65Dn mice has yielded an embarrassment of riches with more than a dozen pharmaceuticals or nutriceuticals reporting promising effects (*Fernandez and Reeves, 2015*). For example, the recent clinical trial by Roche of the GABA-$\alpha$5 antagonist RG1662 gave serious weight to experiments in mouse models showing that normalization of the balance of inhibitory and excitatory inputs to hippocampus could improve several hippocampal-based behavioral paradigms as well as long term potentiation (LTP), all of which are impaired in Ts65Dn and other DS models (*Braudeau et al., 2011*; *Siarey et al., 1997*; *Kleschevnikov et al., 2004*; *Hart et al., 2017*). Improvements with RG1662 are dramatic in mice, but the human trial was terminated in Phase IIa due to lack of efficacy in a long term treatment paradigm (https://clinicaltrials.gov/ct2/show/NCT02024789). The degree to which the failure to translate a preclinical result to human clinical trials relates to the model itself is unknown. However, it seems reasonable to expect that a better genetic representation of trisomy 21 will produce more relevant results.

The current machine-based annotation of HSA21 (GRCh38.p12, BioMart-Ensembl, May 2019) shows that there are 17 and 213 protein coding genes (PCGs) mapped to the short (HSA21p) and long arms of HSA21 (HSA21q), respectively. However, 15 of 17 HSA21p PCGs are identified by BLASTN as base-perfect copies/duplications of HSA21q, and BAGE2 and TPTE are the only unique HSA21 PCGs in HSA21p (*Table 1*). In a detailed review of the annotation, Gardiner and colleagues point out that true duplications on HSA21p would show at least some sequence divergence and suggest that sequence assembly problems account for most or all gene assignments to HSA21p (*Gupta et al., 2016*). Gene annotations for all non-humanized mouse DS models are based on HSA21q orthologs found on mouse chromosomes (MMU) 16, 17 and 10 (*Table 1—source data 1A*).

A transchromosomic (Tc) HSA21 mouse line, Tc1, is the only existing DS mouse model carrying HSA21. The HSA21 was irradiated in the process of creating Tc1 mice, and more than 50 PCGs on HSA21q are disrupted by rearrangement, deletion or duplication (*Gribble et al., 2013*; *O'Doherty et al., 2005*; *Choong et al., 2015*). Tc1 thus includes 158 of 213 functionally trisomic HSA21q PCGs (*Table 1—source data 1B*). An important limitation of Tc1 is that all mice are mosaic for HSA21. Mosaicism makes each Tc1 mouse unique from every other mouse, because the developmental trajectories dependent on cell-cell interactions are different in every individual (*Besson, 2005*;

**Table 1.** PCG content of HSA21p vs. HSA21q.

| HSA21p | gene start | HSA21q Paralog | gene start |
| --- | --- | --- | --- |
| FP565260.4 | 5011799 | DNMT3L | 44246339 |
| FP565260.3 | 5022493 | ICOSLG | 44217014 |
| GATD3B | 5079294 | GATD3A | 44133610 |
| FP565260.2 | 5116343 | GATD3A | 44133610 |
| FP565260.1 | 5130871 | PWP2 | 44107373 |
| FP565260.6 | 5155499 | TRAPPC10 | 44012309 |
| CU639417.1 | 5972924 | H2BFS | 43565189 |
| SIK1B | 6111134 | SIK1 | 43414483 |
| CBSL | 6444869 | CBS | 43053191 |
| U2AF1L5 | 6484623 | U2AF1 | 43092956 |
| CRYAA2 | 6560714 | CRYAA | 43169008 |
| SMIM11B | 7744962 | SMIM11A | 34375480 |
| FAM243B | 7768884 | FAM243A | 34400317 |
| SMIM34B | 7784482 | SMIM34A | 34418715 |
| KCNE1B | 7816675 | KCNE1 | 34446688 |
| BAGE2 | 10413477 | | |
| TPTE | 10521553 | | |

The online version of this article includes the following source data for Table 1:

Source data 1. The analysis of HSA21 genes.PCG content of HSA21p vs. HSA21q.

*Roper and Reeves, 2006*). Mosaicism in Tc1 is not an isolated case, as the retention rates of other human chromosomes and human artificial chromosomes (HACs) vary widely in different mouse tissues (*Shinohara et al., 2000*; *Tomizuka et al., 1997*; *Takiguchi et al., 2014*). The centromere is a multi-function regulator of genome stability and linked to chromosomal instability found in the inter-specific hybrids (*Lerit and Poulton, 2016*; *Metcalfe et al., 2007*). Here, the use of mouse artificial chromosome vectors (MACs) which are freely segregating and capable of carrying Mb-sized genomic segments in mice (*Takiguchi et al., 2014*; *Kazuki et al., 2019*) supported the transfer of a nearly complete copy of HSA21q into the mouse using a MAC-mediated genomic transfer (MMGT) system. The new humanized mouse model of DS, TcMAC21, carries a substantially intact, freely segregating chromosome, and recapitulates several features of DS.

## Results

### Construction and whole genome sequencing (WGS) of TcMAC21 mice

The MAC1 vector contains the centromeric region of MMU11 (*Takiguchi et al., 2014*). We previously constructed hybrid A9 cells containing a copy of HSA21 (*Inoue et al., 2001*), which was then moved into DT40 cells by microcell-mediated chromosome transfer (MMCT) (*Tomizuka et al., 1997*; *Kazuki et al., 2011*). A loxP site was introduced into the NC_000021.9 locus (from 13,021,348 bp to 13,028,858 bp) of HSA21 to create the 'HSA21-loxP' chromosome, which was transferred by MMCT into Chinese hamster ovary (CHO) cells containing the MAC1 vector (*Figure 1A*). Reciprocal translocation between the MAC1 vector and HSA21-loxP was induced by a transient Cre expression to produce the target hybrid chromosome, 'HSA21q-MAC', containing the centromere plus 2.5 kb of the peri-centromeric region of MMU11 (without producing any transcripts) and HSA21q (*Figure 1—figure supplement 1A*). The recombinant clones were selected using HAT, and 16 out of the 18 drug-resistant clones were PCR-positive with Cre-loxP recombination-specific primers. Two lines out of the examined eight clones were confirmed by FISH to contain HSA21q-MAC (*Figure 1—figure supplement 1B–C*). The HSA21q-MAC from each line was introduced into TT2F female mouse ES cell

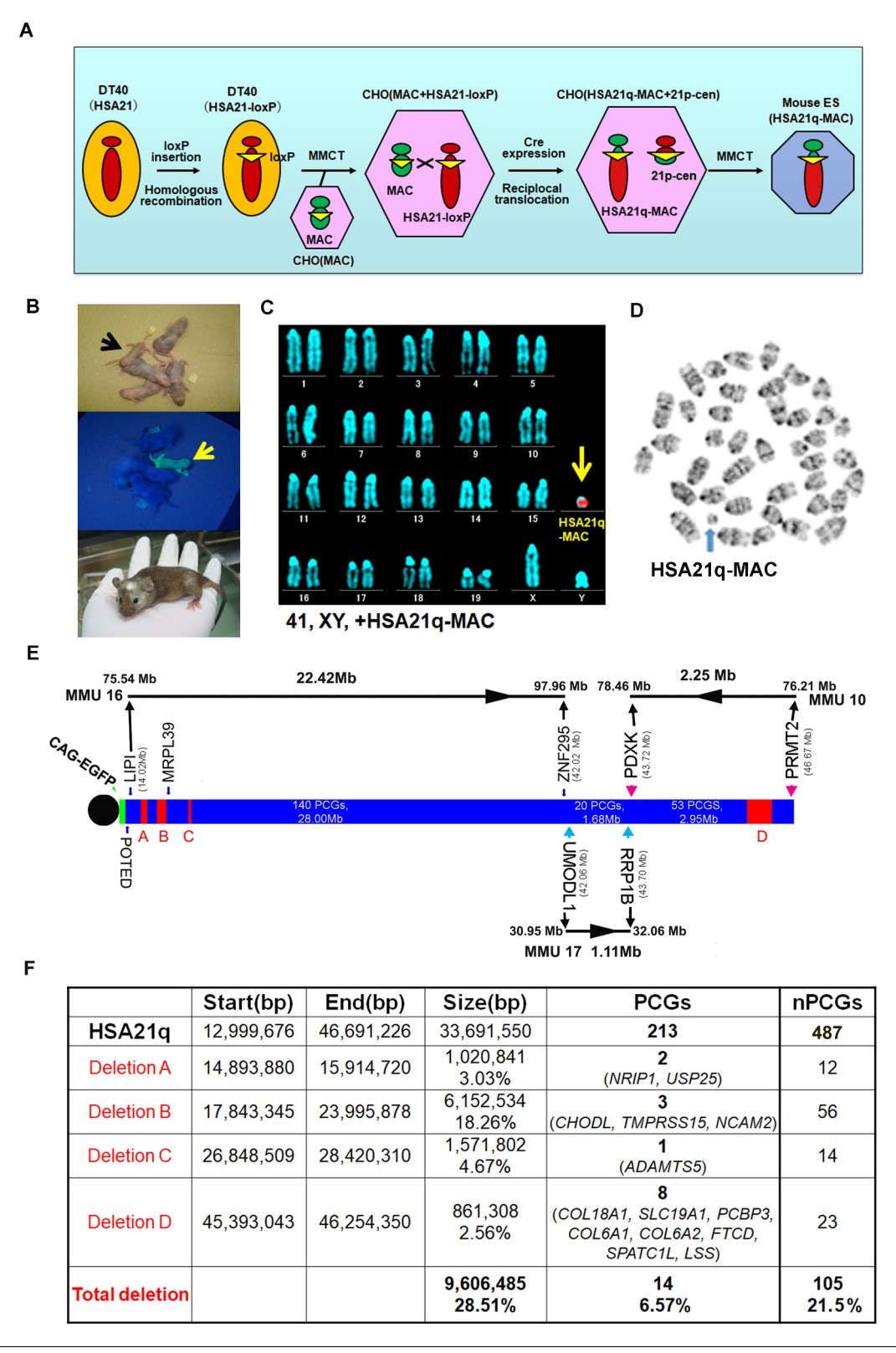

**Figure 1.** Construction of TcMAC21 mice (HSA21q-MAC). (**A**) Schematic diagram of HSA21q-MAC construction. (**B**) Chimeric mice obtained via injection of mouse ES cells carrying the HSA21q-MAC. The arrow indicates a GFP-positive, TcMAC21 mouse. (**C**) FISH of TcMAC21 lymphocytes (n = 4 and 20 metaphases analyzed in each sample). Digoxigenin-labeled human COT-1 DNA as FISH probe for HSA21q-MAC detection. (**D**) G-banding based karyotype of TcMAC21 containing the HSA21q-MAC. (**E**) WGS showing the positions of 4 deletions in HSA21q (A, B, C, and D). These are shown

*Figure 1 continued on next page*

*Figure 1 continued*

normalized to PCG numbers, not physical DNA length. The regions of homology with mouse chromosomes 16, 17 and 10 are indicated. (F) Summary of genome positions of deletions and numbers of affected PCGs and non-PCGs on the HSA21q-MAC (based on GRCh38.p12, BioMart-Ensembl, May 2019), and see *Figure 1—source data 1* for details.

The online version of this article includes the following source data and figure supplement(s) for figure 1:

**Source data 1.** WGS of TcMAC21.
**Figure supplement 1.** Construction of TcMAC21 mice (HSA21q-MAC).

lines using MMCT. FISH confirmed that two ES clones contained HSA21q-MAC as a freely segregating chromosome (*Figure 1—figure supplement 1D*). These were used to produce chimeras that had various degrees of coat-color chimerism (*Figure 1B*). Six GFP-positive female chimeras were crossed with ICR males to establish a novel transchromosomic 'TcMAC21' mouse strain and a line stably segregating the HSA21q-MAC was recovered. FISH and G-banding identified the supernumerary artificial chromosome in the TcMAC21 pups (*Figure 1C–D*).

WGS confirmed that the HSA21q in TcMAC21 extends from the loxP site at base pair 13,021,348 to the telomere at 46,691,226. This region contains all annotated HSA21q PCGs from *POTED* to *PRMT2* and all except one non-PCG, AP001464.1. WGS revealed four deletions in the transchromosome (*Figure 1E*). While physical deletions encompassed ~29% of HSA21q, they occurred substantially in PCG-poor regions, and only 14 of 213 HSA21q PCGs were deleted (*Figure 1F* and *Figure 1—source data 1B–C*). There were 91 single nucleotide variations (SNVs) in HSA21q identified by WGS, and we confirmed that 90 of 91 were previously reported as native alternative alleles (*Figure 1—source data 1D*). A new SNV was observed in the *SON* gene (HSA21: 33,554,276 (reference (C) and SNV(A), P1682Q). Annotation with SIFT predicts that the *SON* SNV is likely to be tolerated, and RNA-Seq (below) demonstrates the presence of a human *SON* mRNA.

## HSA21 genes are expressed, producing dosage imbalance in TcMAC21 mice

We used RNA-Seq to examine HSA21q gene expression by comparing forebrain transcript levels between TcMAC21 and euploid (Eu) at postnatal day 1 (P1). Neither 49 HSA21q keratin associated protein genes (*KRTAP*s) nor their mouse orthologs were expressed in TcMAC21 or Eu brain. In TcMAC21, 66.5% of HSA21 PCGs and 11.1% of HSA21 non-PCGs had medium or high expression (measured as fragments per kilobase of transcript per million mapped reads (FPKM) >0.5), and 36.5% of HSA21 PCGs and 1.2% of HSA21 non-PCGs had high expression (FPKM >5) (*Figure 2A* and *Figure 2—source data 1A*). The most proximal HSA21q non-PCG and PCG, ANKRD30BP2 (13,038,159–13,067,033, FPKM = 5.3) and *POTED* (13,038,159–13,067,033, FPKM = 2.4), respectively, were detected as expressed, as were the most distal HSA21q non-PCG and PCG, DSTNP1 (46,635,166–46,665,124, FPKM = 6.5) and *PRMT2* (46,635,166–46,665,124, FPKM = 60.2) (*Figure 2B*). Transcripts of HSA21 genes located in deleted regions were not detected. These data further demonstrate that the HSA21q-MAC spans the whole long arm of HSA21, is consistent with deletion mapping from WGS, and indicates that the mouse chromosome background (MAC) does not interfere HSA21 gene transcription.

Most (476 of 487) HSA21 non-PCGs do not have a reported mouse ortholog (*Figure 2—source data 1A*). Four HSA21 microRNAs have mouse orthologs, but transcript levels from either mouse or human microRNAs were extremely low or undetectable (*Figure 2—source data 1B*). There are 160 PCGs on HSA21q (excluding KRTAPs) that have mouse orthologs (*Figure 2—source data 1C*). Of these, 117 HSA21 mouse orthologs were expressed with FPKM ≥1 in Eu mice (*Figure 2C* and *Figure 2—source data 1D*). Overall, expression of these mouse orthologs was not substantially affected by the presence of HSA21q in TcMAC21, as 106 out of the 117 orthologs were expressed at 80–120% of Eu levels in TcMAC21. Two mouse orthologs, *Erg* and *Prdm15*, had reduced expression (<0.8 fold), and nine mouse orthologs showed expression >1.2 fold in TcMAC21.

Transcript levels of HSA21 PCGs were on average about half that of their mouse orthologs in Eu mice as expected for a single copy of the human chromosome, but this HSA21/mouse ortholog expression ratio varied considerably among genes. We calculated a total expression value, the sum of FPKM for an HSA21 PCG plus its mouse ortholog, and compared this value to the mouse ortholog

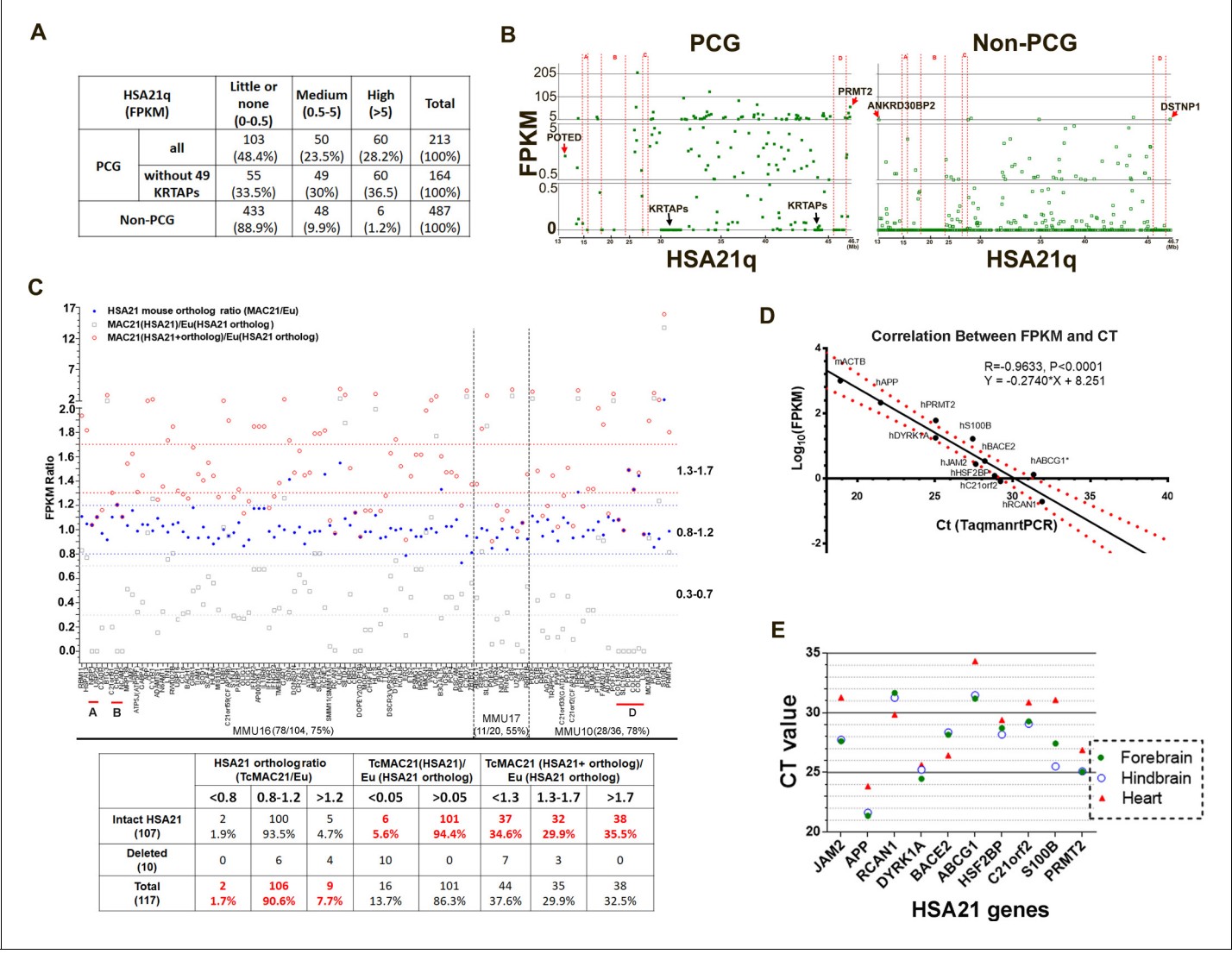

**Figure 2.** HSA21 expression pattern in P1 TcMAC21 brain. (**A**) RNA-Seq summary of HSA21 PCG and non-PCG transcript levels in TcMAC21. (**B**) Transcript levels of individual PCG and non-PCGs across the length of HSA21q in TcMAC21. (**C**) HSA21 dosage imbalance analysis of TcMAC21 among 117 HSA21 mouse orthologs whose FPKM ≥1 in Eu. Three expression values are shown: 1. the FPKM ratio of HSA21 PCG of TcMAC21 to its ortholog of Eu (gray open squares); 2. the FPKM ratio of HSA21 mouse ortholog of TcMAC21 to that of Eu (blue dot); 3. the FPKM ratio of total expression (HSA21 PCG + its mouse ortholog) of TcMAC21 to the HSA21 mouse ortholog of Eu (red circles). The positions of deleted regions are indicated in red. Eu and Ts65Dn littermates, n = 2 per group. See also *Figure 2—source data 1* for expression levels (FPKM) of all HSA21 and its orthologs in Eu and TcMAC21. (**D**) RNA-Seq verification by Taqman RT-PCR. Correlation between CT value from Taqman RT-PCR and $Log_{10}$(FPKM) from RNA-Seq for 10 HSA21q genes and mouse actin (mACTB) in TcMAC21. (**E**) Taqman assay comparing expression of 10 HSA21 genes between forebrain, hindbrain, and heart using the same amount of total RNA. The sample size of Taqman assay in D and E is that n = 2 for TcMAC21 and n = 3 for Eu (negative control). The online version of this article includes the following source data for figure 2:

**Source data 1.** RNA-seq of TcMAC21 and Eu.
**Source data 2.** Effects of HSA21 on gene expression of other mouse chromosomes analyzed by RNA-seq.

expression from Eu (*Figure 2C*). We observed considerable variation around a 1.5-fold average increase in expression. Among intact HSA21 PCGs, 34.6% fell in the low overexpression range (<1.3 fold increased over Eu), 29.9% were in the expected range (1.3–1.7-fold), 35.5% were highly overexpressed (>1.7 fold).

To test effects of HSA21q on gene expression from other mouse chromosomes, we analyzed gene expression changes among 13976 mouse genes (both non-coding and coding) whose

FPKM $\geq$ 1 in Eu (*Figure 2—source data 2*). We found 712 of these genes in TcMAC21 were downregulated (TcMAC21/Eu <0.8) and 1191 genes were up-regulated (TcMAC21/Eu >1.2), indicating that, as in other mouse models and people with trisomy 21, steady state RNA levels are perturbed throughout the genome.

To validate RNA-Seq results, 10 human transcript-specific Taqman probes were used to assess P1 forebrain mRNA. Relative expression quantified by Taqman RT-PCR was significantly correlated with that determined by RNA-Seq (R = 0.96, p<0.0001) (*Figure 2D*). Taqman RT-PCR of different tissues of P1 TcMAC21 mice showed that HSA21 PCG expression levels in forebrain and hindbrain were very similar and frequently different from levels of the same transcript in heart, consistent with tissue-specific regulation of HSA21 genes (*Figure 2E*). Thus, these results indicate that HSA21q is actively transcribed and regulated to produce dosage imbalance in TcMAC21.

## TcMAC21 mice are not mosaic for trisomy

The peri-centromeric GFP on the HSA21q-MAC allows rapid identification of TcMAC21 by illuminating with UV light (Nightsea). To exclude the possibility that GFP-positive mice could carry a distally deleted chromosome due to translocation or chromosome breakage, we validated the model initially by human specific Taqman assays for ten existing TcMAC21-HSA21q genes plus two HSA21 genes that are absent from TcMAC21 as negative controls. Two TcMAC21 and three Eu littermates showed expected expression patterns indicating that the entire chromosome was retained in TcMAC21 (*Figure 3—source data 1*). Twelve additional GFP positive (trisomic) and negative (Eu) pairs were analyzed by Taqman for HSA21q proximal and distal markers (*APP* and *PRMT2*, respectively, *Figure 3— figure supplement 1A*). Including the TcMAC21 mouse used for WGS, we saw 100% concordance of GFP with HSA21q gene expression across the chromosome and zero false positives among fifteen pairs. Thus, GFP appears to be a reliable marker for genotyping TcMAC21.

At a gross level, we saw no evidence of mosaicism in skin, which would appear as patches of nonfluorescent cells when examined for GFP fluorescence (*Figure 3—figure supplement 1B*). Similarly, organs from TcMAC21 mice appeared to be uniformly labeled with GFP (*Figure 3A*), and humanspecific RT-PCR on RNA from nine organs showed the expected levels of expression for the eight HSA21 genes tested (*Figure 3B*). Several tissues were dissociated and analyzed using FISH, whereby the HSA21q-MAC was detected in $\geq$96% of TcMAC21 cells (*Figure 3C–D* and *Figure 3—figure supplement 1C*). Similarly, flow cytometry (FCM) analysis of lymphocytes showed that over 92% of TcMAC21 cells were GFP positive (*Figure 3E*). Immunostaining of parasagittal brain sections with calbindin antibody showed all Purkinje cells (PCs) in TcMAC21 were GFP positive while those in Eu were GFP negative (*Figure 3F*). These stable profiles are consistent with those observed previously in mice carrying the empty MAC1 vector (*Takiguchi et al., 2014*; *Kazuki et al., 2013*). Together, these findings indicate that, if it occurs at all, mosaicism is rare in TcMAC21.

## TcMAC21 mice have abnormalities in heart, brain and skull

We assessed TcMAC21 for phenotypes that occur frequently in DS and other DS mouse models.

### Congenital heart defect (CHD)

CHDs are present in nearly half of newborns with DS and include septal defects, such as the complete AV canal form of atrioventricular septal defect (AVSD), ventricular septal defect (VSD), and atrial septal defect (ASD), plus outflow tract abnormalities, such as double outlet right ventricle (DORV). Several mouse DS models show high rates of CHD, as well (*Williams et al., 2008*; *Lana-Elola et al., 2016*; *Liu et al., 2011*; *Li et al., 2012*). We examined hearts of TcMAC21 mice in a mixed, outbred background by both wet dissection at E18.5 and histology at E14.5. One TcMAC21 heart had both VSD and DORV in wet dissection (*Figure 4A*), while another mouse had AVSD with a small dorsal mesenchymal protrusion (DMP) (*Figure 4B*). We found that 28.6% of TcMAC21 had a structural defect of the heart by consolidating data of both assays (*Figure 4C*). VSD was the predominant malformation, accounting for 21.4%, and AVSD accounting for 2.4%. The 24% frequency of septal defects in TcMAC21 is substantially greater than the 4% observed in Ts65Dn mice but smaller than the 38–55% reported in Tc1 (*O'Doherty et al., 2005*; *Williams et al., 2008*; *Li et al., 2012*; *Dunlevy et al., 2010*). AVSD occurs in 20% of newborns with DS but is rare or absent from DS

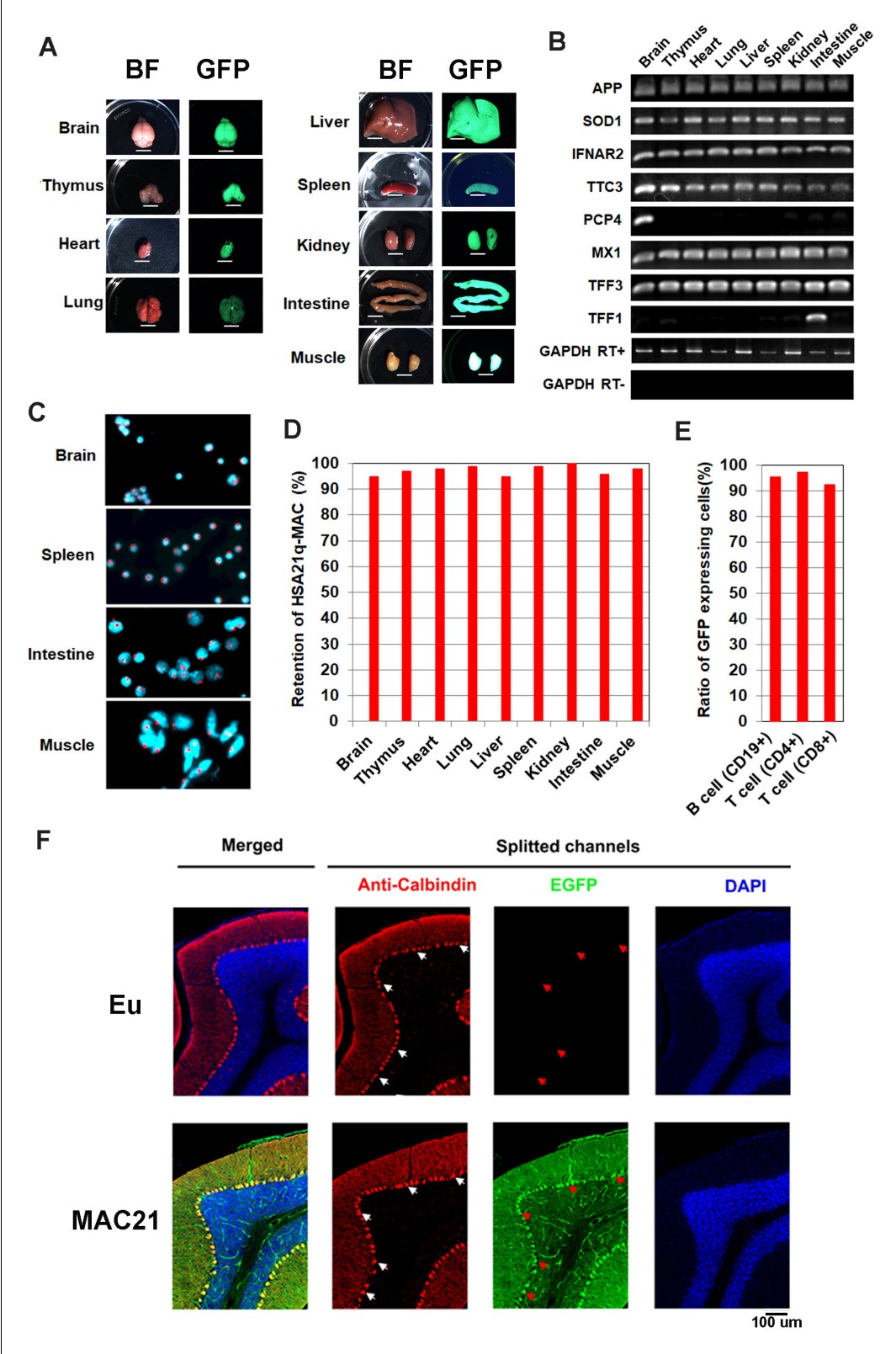

**Figure 3.** Mosaicism analysis in TcMAC21. (**A**) UV light to illuminate GFP of nine organs from TcMAC21 (exposure time is 4 s and 0.2 s in spleen and the other tissues, respectively) (n = 2). BF, bright field. Scale bar (6 mm). (**B**) RT-PCR analyses for HSA21 gene expression in different TcMAC21 tissues (n = 2). (**C**) Representative FISH image of cells from TcMAC21 tissues with digoxigenin-labeled human COT-1 DNA as FISH probe for HSA21q-MAC (red) and with DAPI as a nuclear counterstain (blue). FISH of Eu tissues with human COT-1 showing no signal (*Figure 3—figure supplement 1C*). (**D**)
*Figure 3 continued on next page*

*Figure 3 continued*

Retention rate of the HSA21q-MAC in various TcMAC21 tissues analyzed by FISH (n = 2, 200 interphase cells analyzed in each sample). (**E**) Retention rate of the HSA21q-MAC in three lymphocyte populations analyzed by FCM (n = 2). See *Figure 3—source data 2* for statistical tables used to generate D and E.(**F**) Mosaicism analysis of HSA21q-MAC in Purkinje cells (PC) of TcMAC21 by immunostaining. White arrows indicate cell bodies of randomly selected PC in RFP-channel, and red arrows indicate corresponding locations in GFP-channel. N = 3 per group, one 30 um brain slice per animal, and 100 randomly selected PCs per slice.

The online version of this article includes the following source data and figure supplement(s) for figure 3:

**Source data 1.** HSA21 genes tested by human specific Taqman assay.
**Source data 2.** Data tables for *Figure 3D-E*.
**Figure supplement 1.** Mosaicism analysis.

mouse models except in the presence of additional genetic modifiers (*Li et al., 2012*; *Polk et al., 2015*).

## Brain morphometric phenotypes

Cerebellar hypoplasia is among the few phenotypes present in every person with DS (*Aylward et al., 1997*). Analysis of Ts65Dn identified a deficit in cerebellar granule cells and correctly predicted its occurrence in people with DS (*Baxter et al., 2000*). This deficit is present in several DS mouse models as well (*Olson et al., 2004*; *Dutka et al., 2015*). Ventricle enlargement, which reportedly is associated with neurodegenerative diseases and aneuploidies including DS, was found in previous DS mouse models including Dp(16)1Yey, Dp1Tyb, Ts1Cje and Ts2Cje (*Lana-Elola et al., 2016*; *Raveau et al., 2017*; *Ishihara et al., 2010*). Because ventricle collapse and cerebrospinal fluid (CSF) loss during brain fixation could cause non-representative measurements in MRI (*Figure 4—figure supplement 1*), 3 out of 7 pairs were imaged live. Young adult TcMAC21 (~4.5 month-old) had a slightly larger total brain volume than Eu (487.9 ± 17.6 mm$^3$ vs. 467.3 ± 14.6 mm$^3$, p<0.05) (*Figure 4D*). To analyze configuration changes, lateral ventricles, major subregions of forebrain, interbrain/midbrain, and cerebellum were compared between Eu and TcMAC21. Absolute volumes of lateral ventricle (p<0.003) and interbrain/midbrain including thalamus (p<0.04), hypothalamus (p<0.02) and superior colliculus (p<0.0001) were significantly increased in TcMAC21 (*Figure 4—source data 1*). Total volume of these four structures was small (accounting for ~8.6% of total brain volume in Eu), but the volume increment from them represented ~41% of the volume difference between Eu and TcMAC21. In TcMAC21, the percentage volume of lateral ventricles (p=0.005) and superior colliculus (p<0.0001) were significantly enlarged, and the percentage volume was essentially the same in neocortex (19.63% and 19.26% in Eu and TcMAC21, respectively, p=0.43) but significantly decreased in cerebellum (13.58% and 12.76% in Eu and TcMAC21, respectively, p<0.0005) (*Figure 4—source data 2*). Together, these data indicate that the TcMAC21 brain has altered configurations including enlarged lateral ventricles and superior colliculus, and smaller cerebellum. The impact of trisomy on brain structure in TcMAC21 is somewhat less than that observed in Ts65Dn (*Baxter et al., 2000*).

## Craniofacial skeleton

The close parallels of the impact of trisomy on skull development in individuals with DS and a number of mouse models is well-documented (*Richtsmeier et al., 2000*; *Starbuck et al., 2011*). We used 3D geometric morphometric analysis to compare TcMAC21 and Eu cranial shape based on micro-CT images. Principal component analysis (PCA) showed separation between the two groups (*Figure 4E–F*). In aspects of facial shape change, TcMAC21 mice have relatively more antero-posteriorly retracted and medio-laterally expanded (i.e., short and wide) snouts than do Eu mice. TcMAC21 neurocranium is slightly more 'globular' than that of Eu. These differences are analogous to but somewhat less pronounced than those seen in other DS mouse models, such as Ts65Dn and Dp(16)1Yey (*Singh et al., 2016*; *Starbuck et al., 2014*).

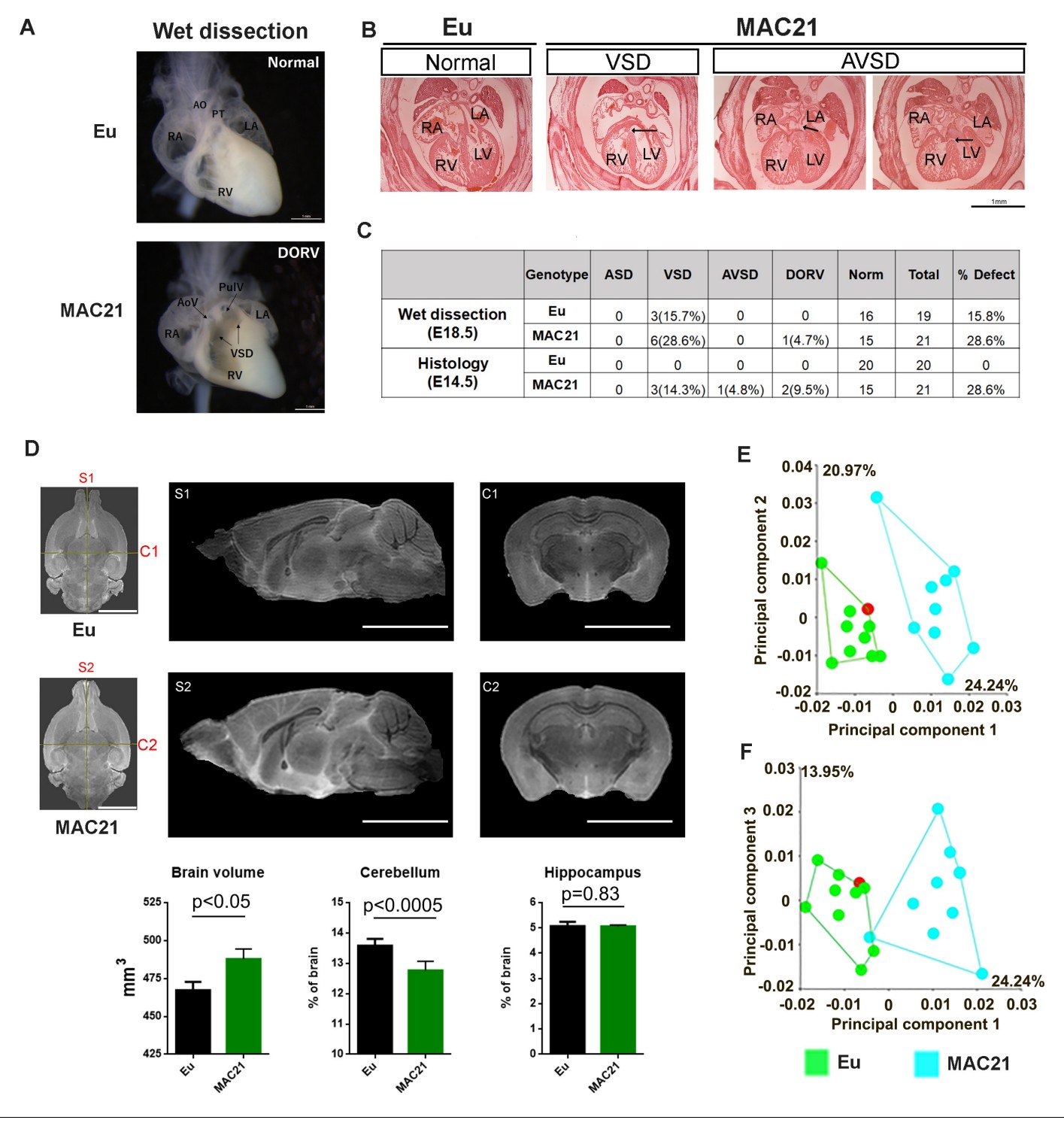

**Figure 4.** Morphological analysis of the heart, brain and skull in TcMAC21. (A–C) CHD analysis. (A) Wet dissection of hearts from E18.5 Eu (top, normal) and TcMAC21(bottom, DORV and VSD (black arrow)), and a small slit-like conus VSD also seen in TcMAC21. Eu (n = 19), TcMAC21 (n = 21). (B) Histology of E14.5 Eu (normal) and two TcMAC21 mice with different heart defects, VSD and AVSD (black arrow indicating locations of defects); AO, aorta; AoV, aortic valve; LA, left atrium; LV, left ventricle; PT, pulmonary trunk; PulV, pulmonary valve; RA, right atrium; RV, right ventricle. Eu (n = 20), TcMAC21 (n = 21). (C) Summary data of CHD analysis. (D) Representative T2-weighted MR images (top, ex vivo) and statistical analysis of whole brain volume, percentage of cerebellum and hippocampus relative to brain; mid-sagittal slices from Eu (S1) and TcMAC21 (S2) and coronal slices from Eu (C1) and TcMAC21 (C2), scale bar (5 mm); n = 7 per group and data are analyzed by two-way ANOVA and expressed as mean ± SEM. (E–F) Results of a

*Figure 4 continued on next page*

*Figure 4 continued*

principal components analysis (PCA) of 3D geometric morphometric analysis of Eu and TcMAC21 cranial shape. PC1 shows a 24.2% separation between Eu and TcMAC21 mice while PC2 captures variation in each genotype. Eu (n = 10), TcMAC21 (n = 9).

The online version of this article includes the following source data and figure supplement(s) for figure 4:

**Source data 1.** Absolute volume of brain structures in TcMAC21 and Eu.
**Source data 2.** Percentage of brain volume in TcMAC21 and Eu.
**Figure supplement 1.** Lateral ventricle in ex vivo and in vivo MRI.

## TcMAC21 mice have various molecular and cellular pathological phenotypes

### Higher APP protein levels without amyloid plaques

Individuals with trisomy 21 invariably display the plaques and tangles characteristic of Alzheimer disease from an early age. These changes are linked to an extra copy of the amyloid precursor protein gene (*APP*) on HSA21 (*Sleegers et al., 2006*; *Prasher et al., 1998*). In P1 forebrain, *APP* was the most highly expressed HSA21 PCG transcript in TcMAC21, and total APP transcripts (HSA21+ its mouse ortholog) of TcMAC21 were about twice as high as the *App* transcript level in Eu (FPKM = 216 in Eu and combined FPKM = 436 in TcMAC21). In 15–24 month-old mice, protein levels of total APP (HSA21 + its mouse ortholog) in hippocampus and cortex of TcMAC21 were about twice as high as that of Eu (p<0.001, *Figure 5A*). Both total Aβ40 and Aβ42 levels in brain were significantly increased in TcMAC21, but the Aβ40/Aβ42 ratio was not significantly different from Eu (*Figure 5B*). Despite elevated APP levels, TcMAC21, like other mouse models of DS, did not show amyloid plaque formation by 15–24 months of age (*Figure 5C*).

### Hematological abnormalities

Children with DS have elevated risk of developing acute leukemia and often develop various hematopoietic disorders (*Choi, 2008*). In peripheral blood analyses, platelet counts were significantly increased in TcMAC21 compared to Eu (112.3 ± 7.8 versus 101.2 ± 6.8; p<0.033) (*Figure 5—source data 1*). TcMAC21 exhibited splenomegaly and white pulp hypertrophy by histology (*Figure 5—figure supplement 1*). Using In vitro colony-forming assays, TcMAC21 showed a significantly reduced frequency of granulocyte/macrophage (GM) in bone marrow and reduced colony formation of GM and granulocyte/erythroid/monocyte/megakaryocyte (GEMM) in spleen (*Figure 5D*), consistent with hematopoietic abnormalities reported in other trisomic mouse models (*Birger et al., 2013*).

### Increased chromosomal radiosensitivity of bone marrow cells

Cultured lymphocytes from individuals with DS are reported to show increased sensitivity to X-rays and various chemical compounds that cause DNA damage (*Sasaki and Tonomura, 1969*). Radiation-induced suppression of the clonogenic activity of hematopoietic stem cells is associated with increased DNA double strand breaks (*Wang et al., 2016*). We observed that following X-ray irradiation, bone marrow cells from TcMAC21 had a significantly higher frequency of aberrations and exchanges in chromatids and chromosomes than Eu (*Figure 5E–G*).

## Husbandry and growth of TcMAC21 mice

BDF1 (C57BL/6J (B6) x DBA/2J (D2)).TcMAC21 females are fertile. Four BDF1.TcMAC21 females back crossed to B6 males produced 15 litters comprising 90 pups. Litter size ranged from 2 to 9 with an average of 6 pups, and 48% of offspring were TcMAC21 (*Figure 6—source data 1*). No fertile TcMAC21 males were produced in B6;D2 mixed background. TcMAC21 females took better care of pups than Ts65Dn females as no apparent pup loss between the birth time and weening age was observed.

Previously, we bred selectively for fertile B6C3 (C57BL/6J (B6) x C3H/HeJ (C3)).Ts65Dn males, and established a Ts65Dn subline in which 42% of trisomic males are fertile (*Moore et al., 2010*). The progeny of these trisomic males produced 28% trisomic mice at weaning, compared to 40% trisomic offspring from Ts65Dn females. When euploid males from the fertile trisomic male Ts65Dn line were crossed to TcMAC21, fertile males were produced. As with Ts65Dn, the frequency of

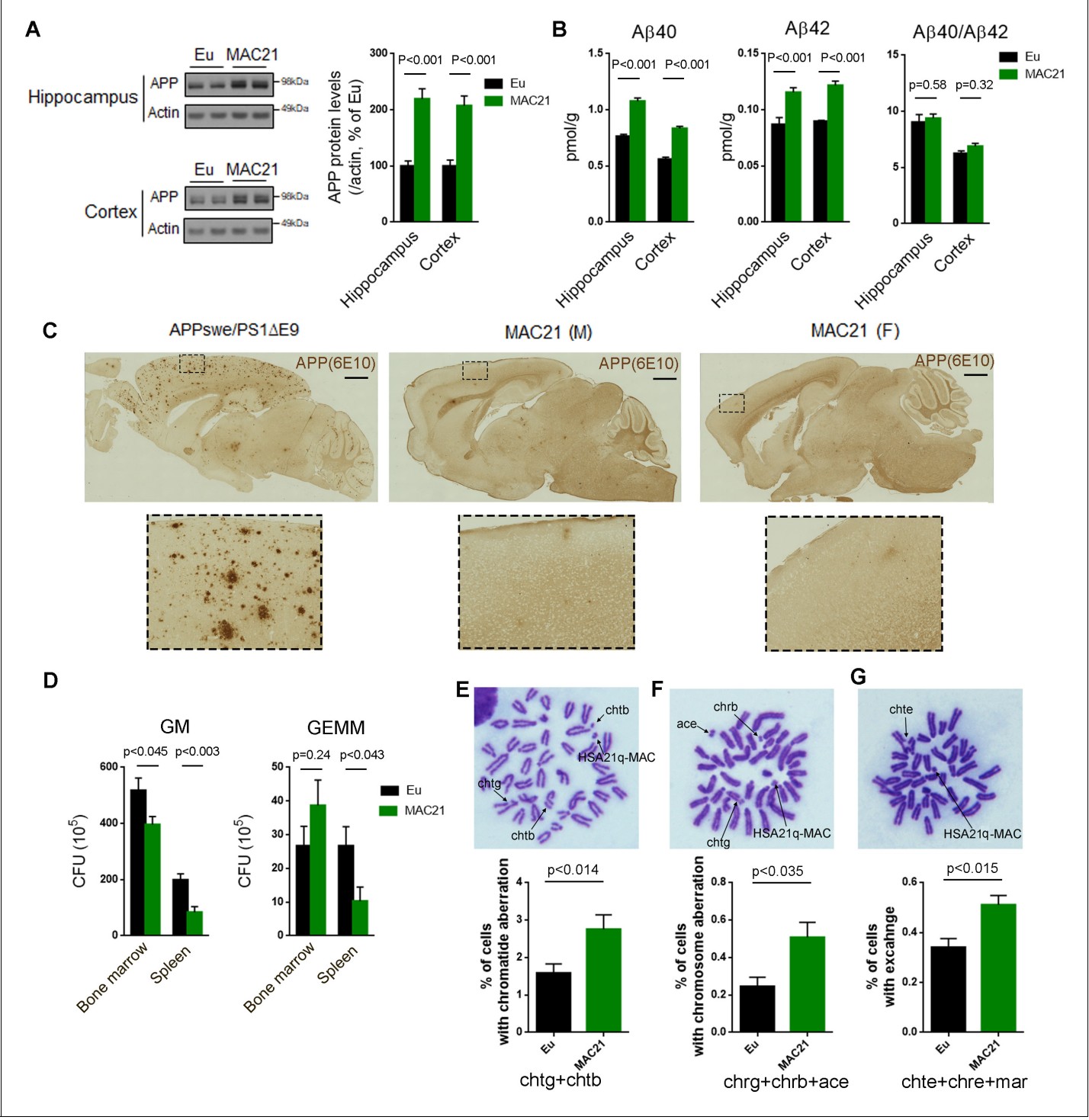

**Figure 5.** Evaluation of DS-like pathology in TcMAC21. (A–C) APP protein levels and amyloid plaques in brains of 15–24 month-old TcMAC21 and Eu (n = 6 per group). (A) western blot of total APP in hippocampus and cortex. (B) ELISA of total Aβ40 and Aβ42 levels in hippocampus and cortex. (C) amyloid plaques visualized by immunostaining with β-amyloid antibody 6E10; *APPswe/PS1ΔE9* mouse is a positive control for plaque formation, scale bar (1 mm). (D) CFU level of GM and GEMM in TcMAC21 spleen and bone marrow (n = 6 per group). GM, granulocyte/macrophage; GEMM, granulocyte/erythroid/monocyte/megakaryocyte. (E–G) Chromatid and chromosome aberrations in bone marrow cells after X-ray irradiation (n = 3 per group). (E) Chromatid aberration. (F) Chromosome aberration. (G) Chromatid and/or chromosome exchange. chtg (chromatid gap); chtb (chromatid break); chrg (chromosome gap); chrb (chromosome break); ace (acentric fragment); chte (chromatid exchange); chre (chromosome exchange); mar (marker chromosome, including dicentric chromosome, ring chromosome, robertsonian translocation, and other abnormal size of chromosome). All data are analyzed by two-tailed t-test and expressed as mean ± SEM.

*Figure 5 continued on next page*

*Figure 5 continued*

The online version of this article includes the following source data and figure supplement(s) for figure 5:

**Source data 1.** Peripheral blood analyses in TcMAC21 and Eu.

**Figure supplement 1.** Hematological phenotypes.

trisomic progeny from TcMAC21 males was 28%, lower than the 50% trisomic offspring from TcMAC21 females.

Individuals with DS show a slower growth rate with delayed developmental milestones, which is reflected in several mouse models of DS (*Hatch-Stein et al., 2016*). We measured the mass of TcMAC21 and Eu mice from P1 through P90 and found that the TcMAC21 cohort was consistently smaller than Eu (*Figure 6A* and *Figure 6—source data 2A*). Both male and female trisomic mice were smaller than their Eu counterparts, and most P90 TcMAC21 females and males weighed less than 20 g and 24 g, respectively. On average, adult TcMAC21 mice were about 25% smaller than Eu at P90.

## TcMAC21 mice have significant learning and memory deficits

Nest construction is a complex task that may require the coordination of various parts of brain including hippocampus, prefrontal cortex, and cerebellum (*Deacon et al., 2002*; *Kolb and Whishaw, 1985*; *DeLorey et al., 2008*). Ts65Dn mice show defects in nesting (*Dutka et al., 2015*; *Salehi et al., 2009*). However, TcMAC21 nest quality was equivalent to that of Eu when tests were performed at 3–4 month-old (*Figure 6B* and *Figure 6—source data 2B*).

To test if TcMAC21 has cognitive deficits, 3-month-old mice (females: Eu (n = 13), TcMAC21 (n = 11); males: Eu (n = 11), TcMAC21 (n = 12)) were tested in open field, visual discrimination water maze test, and Morris water maze (MWM). After MWM, females (Eu (n = 10), TcMAC21 (n = 9)) were further tested in repeated reversal water maze (RRWM), while hippocampal slices from males (Eu (n = 5), TcMAC21 (n = 5)) were analyzed by theta burst stimulation (TBS)-induced long-term potentiation (LTP) (*Figure 6—figure supplement 1*).

First, we used a 30 min novel open field paradigm to assess novelty-induced exploratory activity and anxiety. TcMAC21 exploratory activity was not significantly different from Eu based on total distance traveled (*Figure 6C*), in contrast to hyperactivity in Ts65Dn (*Faizi et al., 2011*). Repeated measures ANOVA of percentage of time spent in the center of the field showed significant effects for HSA21 (p=0.0009) but not for gender (p=0.6) or their interaction (p=0.3) (*Figure 6—source data 2C*), indicating less anxiety in TcMAC21. The differences between Eu and TcMAC21 were shown in male and female separately (*Figure 6D*).

The visual discrimination task showed no significant difference between TcMAC21 and Eu based on both escape latency (p=0.27) and escape distance (p=0.19) (*Figure 7—figure supplement 1* and *Figure 7—source data 1A*), indicating that TcMAC21 has normal visual ability and goal-directed behavior. MWM is evaluated based on the four days of overall performance in acquisition, short delay probe (30 min) and long delay probe (24 hr and 72 hr). Of these three parameters, acquisition is the least sensitive while long delay probe is the most sensitive. There was no significant difference in swimming speed between TcMAC21 and Eu (*Figure 7—figure supplement 2B*). Repeated measures ANOVA showed effects of HSA21, gender, and their interaction in acquisition (escape distance, HSA21: p=0.015, gender: p=0.53, the interaction: p=0.53), short delay probe (HSA21: p<0.002, gender: p=0.55, and the interaction: p=0.59), and long delay probe trials (HSA21: p<0.0001, gender: p=0.01, the interaction: p=0.38) (*Figure 7—figure supplement 2C–E* and *Figure 7—source data 1B*), that is the presence of HSA21 has significant effects on MWM outcomes. Fisher's least significant difference (LSD) post-hoc test showed that in males (Eu (n = 11), TcMAC21 (n = 12)), HSA21 had significant effects on all MWM parameters including acquisition (p<0.05 in escape latency and p<0.027 in escape distance), short delay probe (p<0.009) and long delay probe (p<0.0001), while in females (Eu (n = 10), TcMAC21 (n = 11)), HSA21 effects were relatively less significant than males but still significant overall as p=0.19, p=0.065 and p<0.02 are in acquisition (escape distance), short delay probe and long delay probe, respectively (*Figure 7A*). Together, these data show that poor MWM performance occurs in both female and male TcMAC21, and that Eu and TcMAC21 female

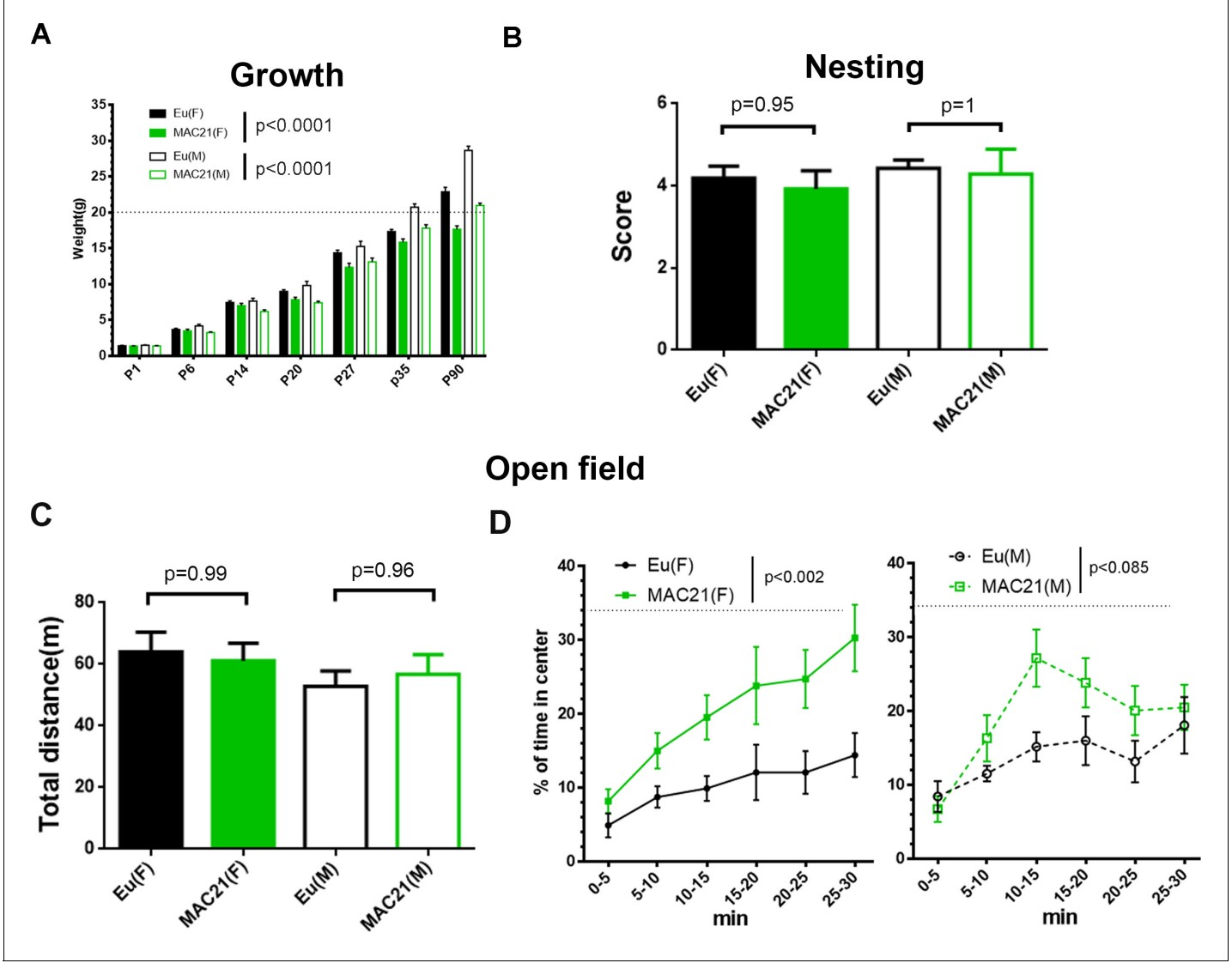

**Figure 6.** Growth profile, nesting and open field in TcMAC21. (**A**) The mass of TcMAC21 and Eu at P1, P6, P14, P20, P27, P35 and P90 (n ≥ 9 per group; two-way ANOVA and Tukey's post-hoc). (**B**) Nesting (female: Eu (n = 19), TcMAC21 (n = 13); male: Eu (n = 7), TcMAC21 (n = 7); one-way ANOVA and Tukey's post-hoc). (**C–D**) 30 min open field (female: Eu (n = 12), TcMAC21 (n = 11); male: Eu (n = 11), TcMAC21 (n = 12)). (**C**) Total travel distance (one-way ANOVA and Tukey's post-hoc). (**D**) Dynamics of % of time spent in the central zone, and dash lines shows that the center zone covers 34% of the whole open field (repeated measures ANOVA with LSD-post-hoc). All data are expressed as mean ± SEM and see *Figure 6—source data 2* for detailed statistical analysis.

The online version of this article includes the following source data and figure supplement(s) for figure 6:

**Source data 1.** Husbandry information for TcMAC21.

**Source data 2.** Statistics for *Figure 6*.

**Figure supplement 1.** The experimental design for behavioral tests and electrophysiology to assess learning and memory in both Eu and TcMAC21.

cohorts may require slightly larger sample size than male cohorts to reach the same statistical significance in MWM.

The forgetting of non-essential memory is an important component of learning and can be assessed by reversal water maze (*Nonaka et al., 2017*; *Brose et al., 2019*), and previous study showed that female but not male APP/PS1 mice showed deficits in this paradigm (*Gallagher et al., 2013*). Immediately following the 72 h probe trials of MWM, females (Eu (n=10), TcMAC21 (n=9)) were tested in RRWM that had the same spatial reference as MWM and consisted of two reversal

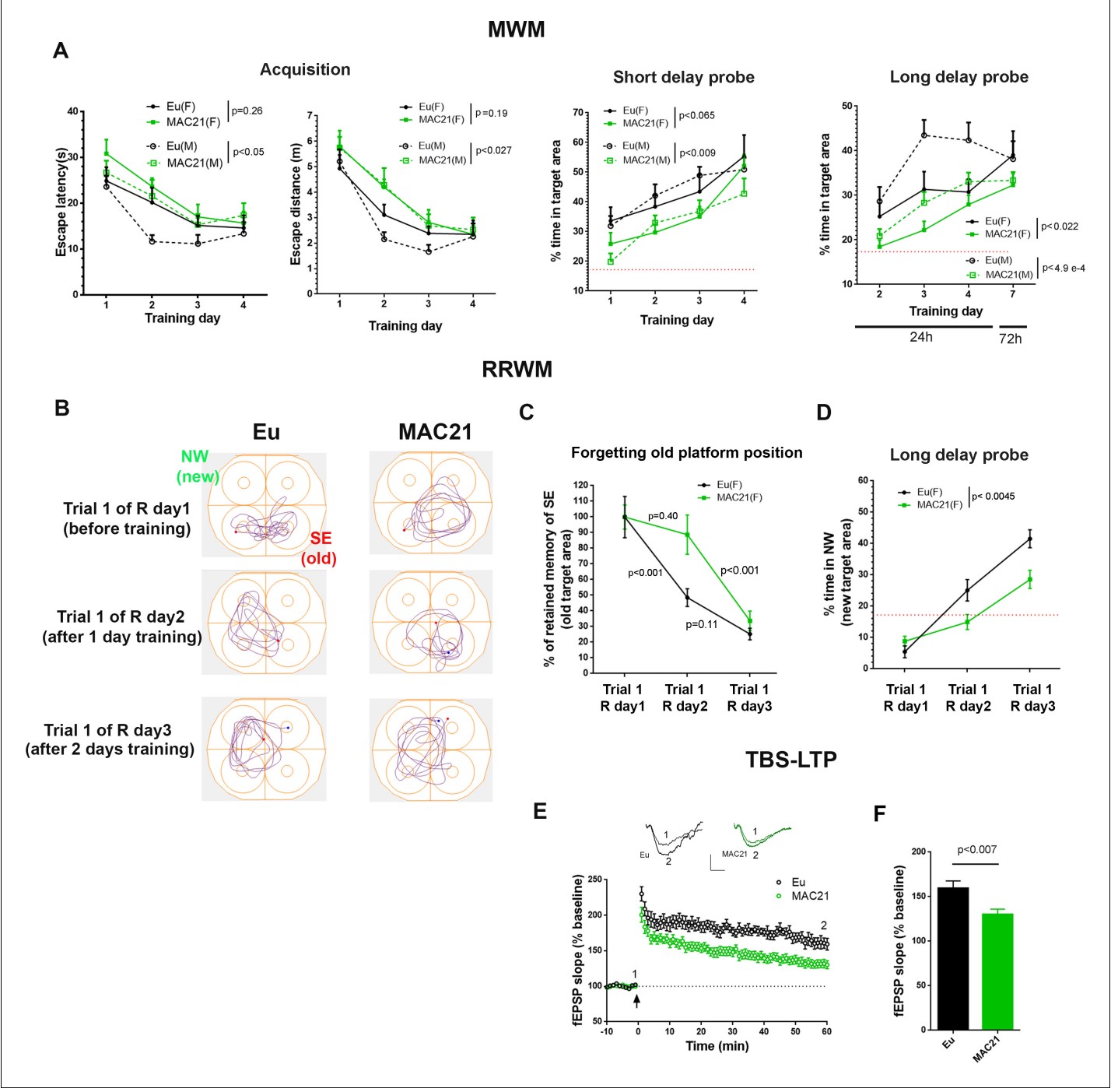

**Figure 7.** Learning and memory deficits in TcMAC21. (**A**) MWM: acquisition trials (left, escape latency and escape distance), short delay probe (middle) and long delay probe (right). Red dash lines indicate 17% chance level (female: Eu (n=10), TcMAC21 (n=11); male: Eu (n=11), TcMAC21 (n=12)). (**B–D**) RRWM (female: Eu (n=8), TcMAC21 (n=9)). (**B**) Representative tracking plots of long delay probe trials of day 1 to day 3 of RRWM to visualize the progress of forgetting old platform SE position and learning new platform NW position. (**C**) Quantitative analysis of forgetting old platform SE position. Before RRWM training, each mouse's "% of retained memory of SE of trial 1 of R day 1" is normalized as 100%. After RRWM day 1 training, "% of retained memory of SE of trial 1 of R day 2" of the mouse = "% of time in SE area in trial 1 of R day 2" / "% of time in SE area in trial 1 of R day 1" × 100%. After RRWM day 2 training, "% of retained memory of SE of trial 1 of R day 3" of the mouse = "% of time in SE area in trial 1 of R day 3" / "% of time in SE area in trial 1 of R day 1" × 100%. (**D**) Long delay probe trials from day 1 to day 3 of RRWM for comparing abilities of learning and memorizing new target platform NW position. (**E–F**) TBS-induced LTP at SC-CA1 synapses (male: Eu (n=5, 13 total slices), TcMAC21 (n=5, 14 total slices)). (**E**) Normalized fEPSP slopes are plotted every 1 min. Sample traces represent fEPSPs taken before TBS stimulation "1" and 60 min after TBS stimulation "2". Arrow indicates LTP induction, and scale bars represent 0.5 mV (vertical), 5 ms (horizontal). (**F**) The amplitude of fEPSP slopes are

*Figure 7 continued on next page*

*Figure 7 continued*

averaged at 60 min after the stimulation. Repeated measures ANOVA with LSD-post hoc test in (**A–D**) and two-tailed t-test in (**E–F**), and data are expressed as mean ± SEM and see *Figure 7—source data 1* for detailed statistical analysis.

The online version of this article includes the following source data and figure supplement(s) for figure 7:

**Source data 1.** Statistics of *Figure 7*.
**Figure supplement 1.** Visual discrimination.
**Figure supplement 2.** MWM.
**Figure supplement 3.** RRWM.

sessions. In reversal session 1 (R day 1 and R day 2) the targeting platform was relocated to NW from SE of MWM, and the platform was relocated to SW in reversal session 2 (R day 3 and R day 4) (*Figure 7—figure supplement 3A*). In RRWM, past platform position such as "SE" of MWM was non-essential memory, while spatial reference was essential memory as it was needed to learn new platform position. We evaluated forgetting by comparing the percent of time spent in the SE target area in long delay probe before RRWM training (trial 1 of R day1, also known as 72 h probe of MWM), with that after 1 day of training (trial 1 of R day2), and with 2 days training (trial 1 of R day3) (*Figure 7B*). To provide a fair quantitative comparison, each animal's "percent of retained memory of SE of trial 1 of R day 1" was normalized as 100%, and "% of retained memory of SE of trial 1 of R day 2" of the mouse was calculated as "% of time in SE area in trial 1 of R day 2" / "% of time in SE area in trial 1 of R day 1" × 100%, and "% of retained memory of SE of R day 3" of the mouse was calculated as "% of time in SE area in trial 1 of R day 3" / "% of time in SE area in trial 1 of R day 1" × 100%. Repeated measures ANOVA and LSD post-hoc test showed that after RRWM day 1 training "% of retained memory of SE" was significantly reduced in Eu (the average reduction >50%, p<0.001) but not in TcMAC21 (the average reduction <15%, p=0.4) (*Figure 7C* and *Figure 7— source data 1C*), indicating that TcMAC21 forgets non-essential memory slower than Eu. In these long delay probe trials, TcMAC21 spent significantly less time in new target NW area than Eu (p<0.0045, *Figure 7D*), indicating deficits of learning and memorizing the new platform position. TcMAC21 performed slightly worse than Eu in acquisition (p<0.05 in escape latency and p=0.15 in escape distance) (*Figure 7—figure supplement 3B*).

Diminished learning and memory ability is strongly correlated with decreased LTP, a measure of synaptic plasticity. After MWM, we assessed TBS-LTP at SC-CA1 synapses on hippocampal slices from Eu and TcMAC21 males (n = 5 per group). TcMAC21 had a significant reduction in LTP compared to Eu littermates (*Figure 7E–F*). Together, behavioral and physiological results showed that TcMAC21 has significant deficits in learning, memory, and synaptic plasticity.

## Discussion

Animal models of DS should first be evaluated on genetic criteria including aneuploidy (an extra freely segregating chromosome), the number of functionally trisomic HSA21 genes or their orthologs at dosage imbalance, the number of functionally trisomic or monosomic non-HSA21 genes, and the status of mosaicism. We compare these criteria for TcMAC21 and previous DS mouse models including Tc1, Ts65Dn, TsCje, Dp1Tyb, Dp(16)1Yey, and the "triple" mouse, Dp(10)1Yey/+;Dp(16)1Yey/+; Dp(17)1Yey/+ (known as DP10/16/17) in *Table 2*. TcMAC21 contains a nearly complete and freely segregating HSA21q as a mouse artificial chromosome and has no detectable mosaicism in a broad spectrum of tissues and cell types. It is the most complete genetic model of DS created to date with 93% of HSA21q PCGs and about 79% of non-PCGs present. At a first approximation, HSA21 genes are transcribed in a tissue appropriate manner and the total transcript increase (human plus mouse ortholog) is broadly at the level that is expected for an additional copy of the affected genes. In contrast, the existing transchromosomic mouse, Tc1, is trisomic for about 75% of HSA21q PCGs and shows significant mosaicism (*O'Doherty et al., 2005*).

Among non-humanized mouse models, only the DP10/16/17 'triple' mouse has a comparable gene content to TcMAC21, containing all 160 HSA21 orthologs (*Yu et al., 2010*). However, the two generation breeding scheme necessary for the three strains carrying the independent duplications is inefficient and live born 'triple' pups are rare (only eight triple DP10/16/17 were recovered among 191 live born pups in a two generation cross in our lab). Further, DP10/16/17 is not formally

**Table 2.** Comparison of TcMAC21 and other mouse models of DS.

| Issues | TcMAC21 | Tc1 | Ts65Dn | Ts1Cje | Dp(16)1Yey/+ or Dp1Tyb | Dp(10)1Yey/+; Dp(16)1Yey/+; Dp(17)1Yey/+ |
|---|---|---|---|---|---|---|
| Mosaicism | No | Yes | No | No | No | No |
| Freely segregating chromosome | Yes | Yes | Yes | No | No | No |
| Trisomic HSA21q PCG/orthologs (% of 213 HSA21q) | 199 (93.4%) | 158 (74.2%) | 110 (51.6%) | 68 (31.9%) | 122 (57.3%) | 180 (40;122;18) (84.5%) |
| Trisomic HSA21q PCG/orthologs, excluding KRTAPs | 150 | 109 | 92 | 68 | 104 | 160 |
| * Trisomic mouse PCGs with no human ortholog | 0 | 0 | 7 | 7 | 10 | 12 |
| † Trisomic or monosomic PCGs that are not HSA21 orthologs | 0 | 0 | Trisomic ~44 MMU17 | Monosomic ~ 7 MMU12 | 0 | 0 |

*Refer to data file S1A for the name and location of each mouse PCG conserved with Hsa21 (MMU10, 16, 17). Gene annotation is based on 'GRCh38, P12' and 'GRCm.P6'.

†The reference of the translocation breakpoints of Ts65Dn and Ts1Cje is "Duchon A, Raveau M, Chevalier C, Nalesso V, Sharp AJ, Herault Y. Identification of the translocation breakpoints in the Ts65Dn and Ts1Cje mouse lines: relevance for modeling Down syndrome. Mamm Genome. 2011;22(11, 12):674–684.' The number of genes is based on 'GRCm.P6'.

aneuploid, i.e., it does not contain an extra centromere on a freely segregating chromosome. Thus, while valid for studies of gene dosage effects, DP10/16/17 and other duplication models lack the distinguishing pathological feature of 95% of people with complete DS. Finally, HSA21 genes without mouse orthologs are obviously not represented DP10/16/17.

The over-expression of human genes in TcMAC21 results in a number of phenotypes that occur in people with trisomy 21 and in other mouse models of DS, including small size (short stature); dysmorphology of the brain with cerebellar hypoplasia; retrusion of the midface skeleton and mandible; congenital heart defects of septation and outflow tract; elevated expression of the Alzheimer-related *APP* gene and its cleavage products; and deficits in learning and memory and their physiological correlate, LTP. We also report the occurrence in TcMAC21 of hematological abnormalities reminiscent of changes related to increased leukemia in children with DS, as well as increased radiosensitivity of bone marrow cells. These established and new observations of parallels between TcMAC21 mice and outcomes of trisomy 21 provide a first validation of the TcMAC21 model for DS research.

There is no perfect animal model for a human genetic disease, less-so for a complex syndrome. For DS research, what one seeks first in an animal model is a genetic representation of the human condition, but ideal genetic representation is not straightforward in DS models. While a human chromosome 21 is more representative of the genes over-expressed in a person with Down syndrome, more than 50 HSA21 PCGs do not have mouse orthologs, raising the question of how they will function in a genome to which they are foreign. Further, human genes with orthologs may not function in precisely the same manner in a mouse. Stoichiometric differences in expression and effect may be expected. This concern has been an issue in mouse models since the first transgenic models were created using human genes but has not precluded relevant analyses across several decades. The working hypothesis for analysis relevant to DS is that many molecular, cellular, and developmental processes will be affected in a similar manner by dosage imbalance for orthologous genes. End phenotypes will necessarily not be identical in mice and people.

DS research has made impressive gains in the last 25 years, substantially fueled by the development of trisomic mouse models. Before the advent of segmental trisomy models, assessments of gene dosage effects in development and function were often based on single gene transgenic mice and necessarily proceeded on a gene by gene basis. Results were interpreted in the context of the reductionist assumption that the clinical presentation is nothing more than the sum of additive, independent effects of (a subset of) HSA21 genes (see *Epstein, 1986*). A considerably more nuanced view has emerged with the use of trisomic models, showing interactions of HSA21 genes, a transcriptome in which disomic as well as trisomic gene expression is perturbed throughout the genome, and genetic interaction of trisomic genes whose elevated expression potentiates phenotypes that require disomic modifiers (*Li et al., 2016*; *Antonarakis et al., 2020*). Despite this progress,

fundamental questions remain about the most complex genetic insult compatible with survival beyond term. The TcMAC21 model provides the best current representation of gene dosage comparable to that of DS, and thus should prove valuable as a new model for preclinical studies aimed at ameliorating effects of trisomy 21 thereby maximizing opportunities for individuals with trisomy 21 to develop their full potentials.

# Materials and methods

## Key resources table

| Reagent type (species) or resource | Designation | Source or reference | Identifiers | Additional information |
|---|---|---|---|---|
| Strain, strain background (*Mus musculus*) | TcMAC21 | This paper | | See details in *Table 3* |
| Antibody | Anti-APP (mouse monoclonal) | Millipore | MAB348, RRID:AB_94882 | (1:500) |
| Antibody | Anti-β-Amyloid (6E10, mouse monoclonal) | Covance | SIG-39320, RRID:AB_662798 | (1:500) |
| Antibody | Anti-Calbindin (rabbit monoclonal) | Cell signaling | 13176, RRID:AB_2687400 | (1:250) |
| Commercial assay or kit | Human/rat β amyloid (Aβ40) | Wako | 294–64701 | |
| Commercial assay or kit | Human/rat β amyloid (Aβ42) | Wako | 290–62601 | |
| Commercial assay or kit | TaqMam Gene Expression Master Mix | Applied Biosystems | 4369016 | |
| Commercial assay or kit | RNeasy Mini Kit | Qiagen | 74104 | |
| Commercial assay or kit | NEBNext Poly (A) mRNA Magnetic Isolation Module | NEB | E7490 | |
| Commercial assay or kit | NEBNext Ultra II RNA Library Prep Kit | NEB | E7770 | |
| Commercial assay or kit | Lipofectamine 2000 Transfection Reagent | ThermoFisher | 11668027 | |
| Chemical compound, drug | Human COT-1 DNA | ThermoFisher | 15279–011 | |
| Chemical compound, drug | Mouse COT-1 DNA | ThermoFisher | 18440–016 | |
| Chemical compound, drug | Mouse minor satellite DNA | Gift of Dr. Vladimir Larionov (NIH) | | |
| Software, algorithm | MorphoJ | Klingenberg lab | | |
| Software, algorithm | R | R-project | | |
| Software, algorithm | Any-maze | Stoeltingco | | |
| Software, algorithm | GraphPad Prism | GraphPad | | |
| Software, algorithm | STATISTICA | TIBCO | | |

## Animals

All procedures related to animal care and treatment were approved by each local University/Institutional Animal Care and Use Committee and met the guidelines of the National Institute of Health and RIKEN Guide for the Care and Use of Laboratory Animals. TcMAC21 mice were initially

maintained on an outbred background (ICR strain mice). After crossing for eight generations onto BDF1 (C57BL/6J (B6) x DBA/2J (D2)), TcMAC21 were transferred to the Riken Animal Resource (BRC No. RBRC05796 and STOCK Tc (HSA21q-MAC1)). Except as specifically noted phenotypic characterizations of TcMAC21, including histology of CHD, MRI, craniofacial morphology and behavioral tests, were performed on mice (75% B6/25% D2 on average) produced by crossing B6 males with trisomic B6D2 females (see *Table 3* for animal information).

## Chromosome engineering to produce HSA21q-MAC

The HSA21q-MAC was constructed using a previously described Mb-sized gene cloning system with the MAC1 vector (*Kazuki et al., 2019*). A loxP site was inserted at position 13021348–13028858, NC_000021.9 in a Chr.21 in DT40 cells as described previously (*Kazuki et al., 2011*). The modified hChr.21 (HSA21-loxP) was transferred to CHO cells, containing MAC1, via MMCT as described previously (*Tomizuka et al., 1997*). MAC1 contains a centromere from mouse chromosome 11, EGFP flanked by HS4 insulators, PGK-neo, loxP-3'HPRT site, PGK-puro, and telomeres (*Takiguchi et al., 2014*). HPRT-deficient CHO cells (CHO HPRT$^{-/-}$) containing MAC1 were maintained in Ham's F-12 nutrient mixture. Cre-recombinase expression vectors were transfected into CHO cells containing the MAC1 and the HSA21-loxP using Lipofectamine 2000. The cell culture and colony expansion were performed as described previously (*Hoshiya et al., 2009*). The site-specific reciprocal translocation between the MAC1 and the HSA21-loxP was confirmed by PCR (*Table 4*) and FISH analyses. Mouse ES cells were fused with microcells prepared from the donor CHO hybrid cells and selected with G418. Parental mouse ES cell line, TT2F, and derivatives were maintained on the mitomycin-treated mouse embryonic fibroblasts.

## Generation of chimeric mice and Tc mice

Chimeric mice were produced from two ES (HSA21q-MAC) cell lines for the Tc mouse line. Chimera production was completed as described previously (*Tomizuka et al., 1997*). Briefly, ES cells were injected into eight-cell-stage embryos derived from ICR mice (CLEA, Tokyo, Japan) and then transferred into pseudopregnant ICR females. Six chimeric mice showing high coat-color chimerism were used to generate Tc mice in which germline transmission could successfully occur. One female F1 mouse was obtained from the F0 and further mated with ICR male. One male F2 mouse was obtained from the F1 and further mated with ICR male. Since F3 was not obtained from the F2 male by normal mating, we collected round spermatids, elongated spermatids and spermatozoa from the F2 male, which were microinjected into ICR or BDF1 oocytes. A total of 217 embryos were constructed and 161 two-cell embryos were transferred into oviducts of pseudopregnant females, which produced 42 offspring, 19 males and 23 females. Among 42 offspring, 18 (6 males and 12 females) (42%) were GFP positive. The 12 GFP positive F3 females were further used for normal mating.

## FISH and G-banding

FISH was performed using fixed metaphase or interphase spreads. Slides were hybridized with digoxigenin-labeled (Roche, Basel, Switzerland) human COT-1 DNA (ThermoFisher) to detect human chromosomes and biotin-labeled mouse COT-1 (ThermoFisher) or minor satellite DNA (a gift from Dr. Vladimir from NIH) to detect mouse chromosomes, essentially as described previously (*Tomizuka et al., 1997*; *Shinohara et al., 2001*). For Giemsa banding, chromosome spreads were prepared and stained as described (*Davisson and Akeson, 1987*).

## Whole genome and RNA sequencing

### WGS

Tail DNA from a TcMAC21 mouse was purified and sequenced for four different runs using the Illumina NextSeq 500 DNA sequencer (*Figure 1—source data 1A*). After cleaning, the short reads were mapped to whole genome sequences of mouse (GRCm38) and human chromosome 21 (NCBI NC_000021.9) using CLC Genomics Workbench ver. 9.5. A total of 974.89 million reads were mapped to the reference sequences. Among these reads, 4.72 million reads were mapped to HSA21, GRCh38.p13 Primary Assembly. Most of the mapped reads were located between positions 13.0M and 46.7M. The effective depth of coverage was 25.5×. All raw read data of WGS were deposited to DDBJ Sequence Read Archive (DRA) under accession number DRA008337 and

**Table 3.** Experimental animal information.

| | Experiment | Genetic background | Age | Eu | TcMAC21 |
|---|---|---|---|---|---|
| *Figure 1* | WGS | B6;DBA | 1-month-old | Not tested | Male: N = 1 |
| *Figure 2* | RNA-seq | B6;DBA | P1 | Male: N = 2 | Male: N = 2 |
| *Figure 3* | GFP expression | B6;DBA | 1–10 month-old | Male: N = 2 | Male: N = 2 |
| *Figure 3* | Mosaicism analysis by FISH | B6;DBA | 1–10 month-old | Male: N = 1 | Male: N = 2 |
| *Figure 3* | Mosaicism analysis by FCM | ICR;B6;CBA | 10-month-old | Male: N = 1 | Male: N = 2 |
| *Figure 3* | Mosaicism analysis by immunostaining | B6;DBA | 4.5-month-old | Male: N = 3 | Male: N = 3 |
| *Figure 4* | CHD by wet dissection | B6;DBA | E18.5 | N = 19, gender unknown | N = 21, gender unknown |
| *Figure 4* | CHD by histology | B6;DBA | E14.5 | N = 20, gender unknown | N = 20, gender unknown |
| *Figure 4* | Brain MRI | B6;DBA | 4.5-month-old | Male: N = 7 | Male: N = 7 |
| *Figure 4* | Craniofacial morphology by micro-CT | B6;DBA | 4.5-month-old | Male: N = 10 | Male: N = 9 |
| *Figure 5* | Western, ELISA, and amyloid plaque staining | B6;DBA | 15–24 month-old | Male: N = 6 | Male: N = 6 |
| *Figure 5* | Peripheral blood analyses | B6;DBA | 10–12 month-old | Male: N = 6 | Male: N = 5 |
| *Figure 5* | Colony-forming cell assays | ICR;B6;DBA | 10-month-old | Male: N = 6 | Male: N = 6 |
| *Figure 5* | Radiosensitivity test | ICR;B6;DBA | 16–18 week-old | Male: N = 3 | Male: N = 3 |
| *Figure 6* | Nesting | B6;DBA | 3–4 month-old | Female: N = 16; Male: N = 7 | Female: N = 13; Male: N = 7 |
| *Figure 6* | Open field | B6;DBA | 3-month-old | Female: N = 12; Male: N = 11 | Female: N = 11; Male: N = 12 |
| *Figure 7* | Classic MWM | B6;DBA | 3-month-old | Female: N = 10; Male: N = 11 | Female: N = 11; Male: N = 12 |
| *Figure 7* | RRWM | B6;DBA | 3-month-old | Female: N = 8 | Female: N = 9 |
| *Figure 7* | TBS-LTP | B6;DBA | 3.5-month-old | Male: N = 5 | Male: N = 5 |

DRA008342. **RNA-Seq:** RNA was extracted from forebrains of Eu and TcMAC21 at P1. Standard mRNA purification and library preparation were conducted using NEBNext Poly (A) mRNA Magnetic Isolation (E7490) and NEBNext Ultra II RNA Library Prep Kit (E7770). Library quality was assessed via Agilent 2100 Bioanalyzer for DNA High sensitivity DNA chip. The prepared library was sequenced using HiSeq2500 Flowcell with 100 bp paired-end reads, with each sample containing approximately 50–60 million reads. Sequence was assessed with fastqc, and 30 bp were trimmed from each sequence. HSA21 reference was extracted and appended onto the whole mouse genome reference sequence to create the modified reference. Reads were then aligned with TopHat2. Sim4 and Leaff were used for cross-species analysis. Standard DEseq methodology was used for differential gene expression analysis.

## CHD analysis by wet dissection and histology

### Wet dissection
E18.5 mouse fetuses were removed and sacrificed, and hearts were flushed with PBS via the umbilical vein and then fixed in 4% PFA. The hearts were examined for cardiovascular anomalies under a dissecting microscope.

### Histology
E14.5 embryos were collected and fixed in 10% formalin for 48 hr. Tissues were embedded in paraffin, sectioned at 10 μM, and stained with hematoxylin/eosin. The heart was analyzed via dissection stereomicroscope (Nikon SMZ1500, Japan).

**Table 4.** Primer sequences for genomic PCR or RT-PCR analyses.

| | Gene or aim | Primer name (forward) | Forward primer(5'−3') | Primer name (reverse) | Reverse primer(5'−3') | Product size |
|---|---|---|---|---|---|---|
| Genomic PCR | MAC1 | m11 5L | TGACAGAGAGCTTCCTCCTGCC TCTGTA | EGFP-F | CCTGAAGTTCATCTGCACCA | 5.0 kb |
| | Chr.21-loxP | #21CEN < 1 > 2L | AAATGCATCACCATTCTCCCAG TTACCC | PGKr1 | GGAGA TGAGGAAGAGGAGAACA | 4.5 kb |
| | Cre-loxP recombination | TRANSL1 | TGGAGGCCA TAAACAAGAAGAC | TRANSR1 | CCCCTTGACCCAGAAATTCCA | 409 bp |
| | Cre-loxP recombination | kj neo | CATCGCCTTCTATCGCCTTC TTGACG | PGKr1 | GGAGA TGAGGAAGAGGAGAACA | ~600 bp |
| | D21S265 | SHGC-40F | GGGTAAGAAGGTGCTTAATGC TC | SHGC-40R | TGAATATGGGTTCTGGATG TAGTG | 178 bp |
| | APP | SHGC-31514F | CTGGGCAATAGAGCAAGACC | SHGC-31514R | ACCCATATTATCTATGGACAA TTGA | 115 bp |
| | D21S260 | D21S260F | AGCTGTTCATGCTTCCATCT | D21S260R | AGAGCCCAGAATATTGACCC | 270 bp |
| | SOD1 | SHGC-6902F | ATTCTGTGATCTCACTCTCAGG | SHGC-6902R | TCGCGACTAACAATCAAAGT | 133 bp |
| | D21S261 | SHGC-3610F | AACACCTTACCTAAAACAGCA | SHGC-3610R | TGGACCTTTTGATTTTTCCT | 130 bp |
| | AML1 | SHGC-30487F | GTAACCTGGTTAACATAGGG TTTC | SHGC-30487R | GTAGGGGAGGCTAATGGCAT | 150 bp |
| | CBR1 | J04056F | GATCCTCCTGAATGCCTG | J04056R | GTAAATGCCCTTTGGACC | 245 bp |
| | SIM2 | WI-22186F | GGGCCTCATGGTAAGAGTCA | WI-22186R | GAAAAATGTCGGTGGTATC TCC | 250 bp |
| | HLCS | WI-15188F | TTCAGTACCTCCCCAGATGC | WI-15188R | CTTAGTAGTGCAGACC TTTACCCC | 125 bp |
| | TTC3 | WI-19945F | TGGACAAATATAAGGCATG TTCA | WI-19945R | GTCACCTTCCTCTGCCTTTG | 267 bp |
| | D21S394 | D21S394F | GGAGCCGGTTCTTCGAAGG | D21S394R | CAGCGTCCGGAATTCCTGC | 71 bp |
| | D21S336 | D21S336F | TCTGGTTCCCAGGATTGTAA | D21S336R | AGAGTTGCTGTAAGCA TCAAAGT | 350 bp |
| | D21S55 | D21S55F | AGGCTCCTTCACCTCTTGAC | D21S55R | CATCCTCTTTGCATTAGG | 159 bp |
| | GIRK2 | GIRK2F | GTTTGTCTTCAGCTCACC | GIRK2R | CCCAAAATACTACACATCC | 266 bp |
| | ERG | M21535F | AATGGCGTCAGCCTCTCC | M21535R | CAGTTTGCCTTACGAGTGG TAGC | 254 bp |
| | ETS2 | SHGC-11267F | TACCATGCCAATGGTTTA TAAGG | SHGC-11267R | ATGTGACTGGGAACATCTTGC | 177 bp |
| | D21S268 | D21S268F | CAACAGAGTGAGACAGGCTC | D21S268R | TTCCAGGAACCACTACACTG | 213 bp |
| | PCP4 | WI-14954F | GAATTCACTCATCGTAACTTCA TTT | WI-14954R | CCTTGTAGGAAGGTA TAGACAATGG | 126 bp |
| | D21S266 | D21S266F | CACAATGTAGATGTAGCACAG TTAG | D21S266R | TGAGTCTGAAGAAAGGCAAA TGAAG | 166 bp |
| | D21S15 | D21S15F | GAGGATAAACCGATTCACAGC TAGGAATAC | D21S15R | GTGCACGTAATTAATGACCA TGATATTGCT | 218 bp |
| | MX1 | WI-18875F | TGGACTGACGACTTGAGTGC | WI-18875R | CTCATGTGCATCTGAGGGTG | 143 bp |
| | TFF1 | SGC35308F | CAGGGATCTGCCTGCATC | SGC35308R | ATCGATCTCTTTTAA TTTTTAGGCC | 183 bp |
| | PWP2 | SHGC-33273F | GATCTTGACCGGGAAAAGGG | SHGC-33273R | AACAAGTGGCAAAATGCATAC | 150 bp |
| | APECED | A009B16F | AAAATCCTCCCTTTAAGAGC | A009B16R | GGGTGTTAGGTACTGGCT | 118 bp |
| | PFKL | sts-X15573F | AGGGCTTCTGAGGCCAGC | sts-X15573R | AGGGCACTCTGTCCTCCTGC | 239 bp |
| | UBE2G2 | WI-11417F | TTCAACAGTCATTAGGTTCCACC | WI-11417R | GTGAGATCGGGAGAGGGAG | 129 bp |

*Table 4 continued on next page*

*Table 4 continued*

| | Gene or aim | Primer name (forward) | Forward primer(5'—3') | Primer name (reverse) | Reverse primer(5'—3') | Product size |
|---|---|---|---|---|---|---|
| RT-PCR | APP | WI-18826F | ACGTTTGTTTCTTCGTGCCT | WI-18826R | GCCCCGTAAAAGTGCTTACA | 136 bp |
| | SOD1 | SHGC-6902F | ATTCTGTGATCTCACTCTCAGG | SHGC-6902R | TCGCGACTAACAATCAAAGT | 133 bp |
| | IFNAR2 | U29584F | CGAAGTTTCAGTCGGTGAG | U29584R | GGCATTCAGGTTTTATCCC | 181 bp |
| | TTC3 | WI-19945F | TGGACAAATATAAGGCATGTTCA | WI-19945R | GTCACCTTCCTCTGCCTTTG | 267 bp |
| | PCP4 | WI-14954F | GAATTCACTCATCGTAACTTCATTT | WI-14954R | CCTTGTAGGAAGGTATAGACAATGG | 126 bp |
| | MX1 | WI-18875F | TGGACTGACGACTTGAGTGC | WI-18875R | CTCATGTGCATCTGAGGGTG | 143 bp |
| | TFF3 | WI-7267F | GGCTGTGATTGCTGCCAG | WI-7267R | GTGGAGCATGGGACCTTTAT | 124 bp |
| | TFF1 | SGC35308F | CAGGGATCTGCCTGCATC | SGC35308R | ATCGATCTCTTTTAATTTTTAGGCC | 183 bp |
| | GADPH | RPC1 | CCATCTTCCAGGAGCGAGA | RPC2 | TGTCATACCAGGAAATGAGC | 722 bp |

## Brain morphometry by MRI

4.5-month-old TcMAC21 and Eu mice were used for either ex vivo or in vivo MRI (*Zhang et al., 2010*). For the ex vivo scan, 4 pairs of mice were perfused with 4% PFA after PBS and heads were post-fixed for 1 week, then kept in PBS for 3 days. Heads were stored in Fomblin to prevent dehydration during imaging in an 11.7 Tesla scanner (vertical bore, Bruker Biospin, Billerica, MA). 3D T2-weighted images were acquired on an 11.7 Tesla Bruker scanner (Bruker Biospin, Billerica, MA, USA) with the resolution = 0.08 mm x 0.08 mm x 0.08 mm. For the in vivo scan, 3 pairs of mice were anesthetized with isoflurane and monitored with a physiological monitoring system during imaging in 9.4 Tesla scanner (Bruker Biospin, Billerica, MA, USA). 3D T2-weighted images were acquired at resolution = 0.1 mm x 0.1 mm x 0.25 mm. For analysis, both ex vivo and in vivo images were first aligned to the template image using automated image registration software (Diffeomap, www.mristudio.org) and adjusted to an isotropic resolution of 0.0625 mm $\times$ 0.0625 mm $\times$ 0.0625 mm.

## Craniofacial morphology by micro-CT

Nine 3D anatomical facial landmarks were collected on micro-CT of 9 TcMAC21 and 10 Eu at age of 4.5 month old. Each specimen's landmark configuration was superimposed using the generalized Procrustes analysis (GPA). This method extracts shape coordinates from the original specimen landmark configurations by translating, scaling, and rotating the data and subsequently yields a measure for size called centroid size (*Dryden and Mardia, 1992*). To examine cranial shape variation in the sample we used principal component analysis (PCA). All analyses were conducted in MorphoJ and software R.

## Western blot, ELISA, and amyloid plaque staining

15–24 month-old Eu and TcMAC21 mice (n = 6 each) were perfused with PBS, then a half brain was used to make lysate for western blot and ELISA, and the other half was fixed in 4% PFA. WesternBlot and ELISA: cortex and hippocampus were removed and lysed with RIPA buffer. Protein extracts were separated by 4–12% SDS-PAGE, transferred to PVDF membranes and then probed with APP antibody (Millipore, MAB348) and beta actin. For ELISA, the above lysate was spun at 16,000 g for 30 min at 4°C. Supernatant was analyzed to determine Aβ levels by human/rat β amyloid ELISA kit from Wako (Aβ40: Cat# 294–64701; Aβ42: Cat# 290–62601). Amyloid plaque staining: mouse brains were fixed by immersion in 4% PFA, embedded in paraffin, and sectioned at 5 μm. Sections were deparaffinized and protein antigenicity was unmasked, and then the endogenous peroxidase activity was inhibited with 1.5% hydrogen peroxide. Sections were incubated with mouse anti-β-amyloid (Covance, Cat# SIG-39320), biotinylated goat anti-mouse IgG, and Avidin/Biotin mixture and then developed in 3,3'-Diaminobenzidine (DAB).

## Peripheral blood analyses

10–12 month-old mice were bled from the retro-orbital plexus under general anesthesia and euthanized for further analyses. Hematopoietic indices were measured with a hemocytometer (Nihon Koden, Japan).

## Colony-forming cell assays

Bone marrow and splenic mononucleated cells were incubated in duplicate at cell concentrations of $2 \times 10^4$ and $2 \times 10^5$/mL, respectively, in MethoCult-M3434 semisolid culture medium (STEMCELL TECHNOLOGIES). Colonies were scored on day 3 for erythroid colony-forming units (CFU-Es) and on day 8 for granulocyte/erythroid/monocyte/megakaryocyte colony-forming units (CFU-GEMMs), granulocyte/monocyte colony-forming units (CFU-GMs), and monocyte colony-forming units (CFU-Ms). For the megakaryocyte colony-forming units (CFU-Megs) assays, $5 \times 10^4$/mL bone marrow and $5 \times 10^5$/mL spleen cells were cultured with MegaCult-C Kit (STEMCELL TECHNOLOGIES) for 9 days, and colonies were stained for acetylcholine esterase (AChE) in accordance with manufacturer's recommendation.

## Radiosensitivity test

TcMAC21 and Eu littermates at 16–18 weeks old received whole body X-ray irradiation at the rate of 0.25 Gy/min for 12 min using MX-160Lab (mediXtec Japan Corporation). After 8 hr, irradiated mice were administered colcemid (1.5 mg/kg) (Demecolcine, Sigma-Aldrich, USA) by intraperitoneal injection. After 3 hr, bone marrow cells were harvested from tibia and femurs of both legs and then fixed in Carnoy's solution. Chromosome spreads were prepared by dropping the cell suspension on a grass slide and stained with 5% Giemsa solution (Merck Millipore, Germany) for 15 min. The analyses for chromosome aberrations were performed using the stained metaphase spreads and classification of chromosome aberrations according to *Shaffer et al., 2009*. For each mouse, >30 mitotic cells were analyzed, and aberration rate was expressed as % of total cell analyzed.

## Behavioral tests

3-month-old mice (females: Eu = 13, TcMAC21 = 11; males: Eu = 11, TcMAC21 = 12) were used for behavioral tests in the sequence, open field, visual discrimination water maze test, MWM, and RRWM. The ANY-maze tracking system (Stoelting Co.) was used to collect data. Open field: was performed after three days of handling in the same room consisting of indirect diffusing light (~150 lux) for all animals. The whole arena size was 37 cm X 37 cm, and the center area (21.6 cm X 21.6 cm) was 34% of the whole arena. Distance traveled and percentage of time spent in center were analyzed in 30 min and 5-min-bins. One female mouse was excluded because of bad tracking, which left (female: Eu (n = 12), TcMAC21 (n = 11); male: Eu (n = 11), TcMAC21 (n = 12)) for statistical analysis. Visual discrimination (VD): the day before VD, all mice were pre-trained to climb and stay on a submerged platform (10 cm X 10 cm) in a small clear water pool (45 cm diameter) for five trials (*Chow et al., 2010*). Non-toxic white tempera paint was used to make the platform invisible in a water tank 126 cm in diameter. There was no spatial cue, but the location of the platform was made visible by attaching a black extension, 4 cm above water surface. The platform position was changed every two trials from W to E to S for a total six trials. Classic MWM: for all animals, the same spatial cues were used and MWM was performed for four days (each day had 10 trials, which included eight acquisition trials plus two probe trials of short- (30 min) and long-delay (24 hr). Trial 5 of MWM Day one was a probe trial drill and no data from the trial was used for analysis: the platform was lowered to a position that mice were not able to climb onto, and mice were only allowed to swim for 10 s, and then a tester raised the platform and guided mice to the platform. A longest delay probe trial was conducted 72 hr after the fourth training day. The target area was defined as a circle inscribed in the platform quadrant, covering ~17% of water maze tank. The platform remained in the same position in 'SE quadrant' during the MWM test, with the water temperature at 22 ± 2℃. The platform was hidden ~1.8 cm below the water surface and 60 s was the maximum time allowed in acquisition trials, and if a mouse did not find the platform, the tester would visually or manually guide it to the platform. For the probe trials, the platform was lowered to a position that mice were not able to climb onto for 30–40 s. At the end of probe trials, the collapsed platform was raised to the same position used in the acquisition trial and the tester guided the mouse to the platform, which helped

maintain the same response-reinforcement contingency of the acquisition. If a mouse continually failed to follow the tester's guidance to reach the platform, it was excluded from analysis. Three female mice were excluded by this standard, which left these mice (female: Eu (n = 10), TcMAC21 (n = 11); male: Eu (n = 11), TcMAC21 (n = 12)) for statistical analysis. RRWM: following the classic MWM, 9 TcMAC21 and 10 Eu female mice were tested in RRWM without changing any spatial cues. RRWM consisted of two reversal learning tests in which the platform was first relocated to NW from SE for two days and then relocated to SW for another two days. Trial 1 of reversal day one was the same as the 72 hr delay probe trial in the classic MWM. Each day had 10 trials including eight acquisition trials and two probe trials for short delay (30 min) and long delay (24 hr). Excluding 2 Eu female mice that failed to follow tester's guidance, 9 TcMAC21 and 8 Eu female mice were analyzed for RRWM. ***Nesting:*** Both female (Eu (n = 19), TcMAC21 (n = 13)) and male (Eu (n = 7), TcMAC21 (n = 7)) mice were tested. 3–4 month-old TcMAC21 mice and their littermates were singly housed with an intact compressed cotton nesting pad in a new cage for 24 hr. The mouse was removed, and the cage was photographed. An observer blinded to the genotypes of mice scored 'nesting quality': 1,<50% of nesting square shredded but not organized; 2,<50% of nesting square shredded and organized or 50–99% of nesting square shredded but not organized; 3, 50–99% of nesting square shredded and organized or 100% of nesting square shredded but not organized; 4, 100% of nesting square shredded and organized into a large nest that covers less than half of the area of the cage; 5, 100% of nesting square shredded and organized into a compact nest that covers less than a quarter of the area of the cage; or 6, 100% of nesting square shredded and organized into a small nest with rounded edges and a 'donut hole' center.

## Electrophysiology

Following behavioral tests, 5 pairs of Eu and TcMAC21 male mice (3–4 months old) were deeply anesthetized with inhaled isoflurane and then perfused with ice-cold oxygenated cutting solution containing (in mM): 110 choline chloride, 7 $MgCl_2$, 2.5 KCl, 0.5 $CaCl_2$, 1.3 $NaH_2PO_4$, 25 $NaHCO_3$, 20 glucose, saturated with 95% $O_2$% and 5% $CO_2$. Transverse hippocampal slices (400 um) were cut using a vibratome (VT-1200S, Leica) and transferred to artificial cerebrospinal fluid (aCSF) containing (in mM): 125 NaCl, 2.5 KCl, 2.5 $CaCl_2$, 1.3 $MgCl_2$, 1.3 $NaH_2PO_4$, 25 $NaHCO_3$, 10 glucose, saturated with 95% $O_2$% and 5% $CO_2$. Slices were to recover for 40 min at 32°C and then at room temperature for at least 2 hr before recording. Picrotoxin (100 μM) was added to block inhibitory transmission. Slices were transferred to the recording chamber and perfused continuously with aCSF (flow rate at 2–3 mL/min) at room temperature. A cut between CA3 and CA1 was made to minimize recurrent activity during recording. A concentric bipolar electrode (World Precision Instruments) was placed in the middle of CA1 stratum radiatum to stimulate Schaffer collateral. Field EPSPs (fEPSPs) from the CA1 neurons were recorded with a glass pipette (2–3 MΩ) filled with aCSF. Constant current pulses (70–100 μA, 100 μs) were delivered at 0.033 Hz by a STG 400 stimulator. The stimulus intensity was adjusted to evoke 40–50% of the maximal response. LTP was induced by theta burst stimulation (TBS) consisting of a single train of 5 bursts at 5 Hz, and each burst contained 4 pulses at 100 Hz. The recording and data analysis were performed by investigators blinded to mouse genotype.

## Statistical analyses

For each experiment, we stated statistical information including the exact sample size, statistical tests, and the exact p-values in each figure or its legend. Unless otherwise noted, data were expressed as mean ± SEM (the standard error of the mean). For behavioral tests, we provided the details of statistical results as *Figure 6—source data 2* and *Figure 7—source data 1*.

## Acknowledgements

We thank Toko Kurosaki, Yukako Sumida, Hiromichi Kohno, Masami Morimura, Kei Yoshida, Eri Kaneda, Akiko Ashiba, Dr. Kazuomi Nakamura, Rina Ohnishi, Yuwna Yakura, Etsuya Ueno, Dr. Yuichi Iida, Dr. Yoshiteru Kai, Motoshi Kimura, Chie Ishihara, Kiyoko Kawakami, and Chiga Igawa at Tottori University for their technical assistance; as well as Dr. Hiroyuki Kugoh, Dr. Masaharu Hiratsuka, Dr. Tetsuya Ohbayashi, Dr. Hiroyuki Satofuka, and Dr. Takahito Ohira at Tottori University, Dr. Ohmiya Yoshihiro at Health Research Institute, National Institute of Advanced Industrial Science and Technology (AIST), and Dr. Shigeharu Wakana, Dr. Tamio Furuse, Dr. Ikuko Yamada at Technology and

Development Team for Mouse Phenotype Analysis Japan Mouse Clinic, RIKEN BioResource Center (BRC) for critical discussions. This research was partly performed at the Tottori Bio Frontier managed by Tottori prefecture. The work was supported in part by JST CREST Grant Number JPMJCR18S4, Japan (YK), JSPS KAKENHI Grant Number 25221308 (MO), Public Health Grants R01HD038384 and R21HD098540 (RHR). Opinions expressed in this article are the authors' own and do not necessarily reflect the view of the National Institutes of Health, the Department of Health and Human Services, or the United States government.

## Additional information

### Funding

| Funder | Grant reference number | Author |
| --- | --- | --- |
| Eunice Kennedy Shriver National Institute of Child Health and Human Development | R21HD098540-01 | Roger H Reeves |
| Eunice Kennedy Shriver National Institute of Child Health and Human Development | R01HD038384 | Roger H Reeves |
| Japan Society for the Promotion of Science | 25221308 | Mitsuo Oshimura |
| Core Research for Evolutional Science and Technology | JPMJCR18S4 | Yasuhiro Kazuki |

The funders had no role in study design, data collection and interpretation, or the decision to submit the work for publication.

### Author contributions

Yasuhiro Kazuki, Roger H Reeves, Conceptualization, Resources, Data curation, Software, Formal analysis, Supervision, Funding acquisition, Validation, Investigation, Visualization, Methodology, Writing - original draft, Project administration, Writing - review and editing; Feng J Gao, Conceptualization, Data curation, Software, Formal analysis, Supervision, Validation, Investigation, Visualization, Methodology, Writing - original draft, Project administration, Writing - review and editing; Yicong Li, Data curation, Investigation; Anna J Moyer, Benjamin Devenney, Kei Hiramatsu, Sachiko Miyagawa-Tomita, Satoshi Abe, Kanako Kazuki, Naoyo Kajitani, Narumi Uno, Shoko Takehara, Masato Takiguchi, Miho Yamakawa, Atsushi Hasegawa, Ritsuko Shimizu, Satoko Matsukura, Naohiro Noda, Narumi Ogonuki, Kimiko Inoue, Shogo Matoba, Atsuo Ogura, Liliana D Florea, Alena Savonenko, Meifang Xiao, Dan Wu, Denise AS Batista, Junhua Yang, Zhaozhu Qiu, Nandini Singh, Joan T Richtsmeier, Takashi Takeuchi, Investigation; Mitsuo Oshimura, Conceptualization, Resources, Funding acquisition, Data curation, Supervision, Investigation, Project administration

### Author ORCIDs

Yasuhiro Kazuki  https://orcid.org/0000-0003-4818-4710
Feng J Gao  https://orcid.org/0000-0002-5548-5055
Atsuo Ogura  http://orcid.org/0000-0003-0447-1988
Joan T Richtsmeier  http://orcid.org/0000-0002-0239-5822
Roger H Reeves  https://orcid.org/0000-0002-3581-0850

### Ethics

Animal experimentation: This study was performed in strict accordance with the recommendations in the Guide for the Care and Use of Laboratory Animals of the National Institutes of Health. All animal experiments were approved by the Institutional Animal Care and Use Committee (IACUC) protocols of Johns Hopkins University (#MO18M291), Tottori University (Permit Number: 06-S-102, 08-Y-69, 09-Y-24,11-Y-52, 13-Y-19, 14-Y-23, 15-Y-31, 16-Y-20, 17-Y-28, 19-Y-22, 20-Y-13), RIKEN BioResource Research Center (Permit Number: 08-005, 09-005, 10-005), and Tohoku University (Permit Number: 2013MdA-424).

## Decision letter and Author response

Decision letter https://doi.org/10.7554/eLife.56223.sa1
Author response https://doi.org/10.7554/eLife.56223.sa2

# Additional files

## Supplementary files

• Transparent reporting form

## Data availability

All raw read data of TcMAC21 WGS were deposited to DDBJ Sequence Read Archive (DRA) under accession number DRA008337 and DRA008342.

The following datasets were generated:

| Author(s) | Year | Dataset title | Dataset URL | Database and Identifier |
| --- | --- | --- | --- | --- |
| Matsukura S, Noda N, Kazuki Y | 2019 | Analysis of DS mouse | https://ddbj.nig.ac.jp/DRASearch/submission?acc=DRA008342 | DDBJ Sequence Read Archive (DRA), DRA008342 |
| Matsukura S, Noda N, Kazuki Y | 2019 | Analysis of DS mouse | https://ddbj.nig.ac.jp/DRASearch/submission?acc=DRA008337 | DDBJ Sequence Read Archive (DRA) , DRA008337 |

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
