## [Decision Letter]

**Acceptance summary:**

We are excited about your description of a new mouse model of Down's Syndrome. Unlike previous models, yours has a freely segregating mouse artificial chromosome that is present in all cells and that contains nearly all genes from the long arm of human chromosome 21. Many Down Syndrome-relevant phenotypes are present in this model and its further study should allow for significant insights into Down's Syndrome biology and its associated pathogenic events.

**Decision letter after peer review:**

Thank you for submitting your article "A non-mosaic transchromosomic mouse model of Down syndrome carrying the long arm of human chromosome 21" for consideration by *eLife*. Your article has been reviewed by three peer reviewers, and the evaluation has been overseen by a Reviewing Editor and Huda Zoghbi as the Senior Editor. The following individuals involved in review of your submission have agreed to reveal their identity: Yann Herault (Reviewer #1); William Mobley (Reviewer #2); Tarik Haydar (Reviewer #3).

The reviewers have discussed the reviews with one another and the Reviewing Editor has drafted this decision to help you prepare a revised submission.

As the editors have judged that your manuscript is of interest, but as described below that additional experiments and/or data analysis are required before it is published, we would like to draw your attention to changes in our revision policy that we have made in response to COVID-19 (https://elifesciences.org/articles/57162).

First, because many researchers have temporarily lost access to the labs, we will give authors as much time as they need to submit revised manuscripts.

We are also offering, if you choose, to post the manuscript to bioRxiv (if it is not already there) along with this decision letter and a formal designation that the manuscript is 'in revision at *eLife*'. Please let us know if you would like to pursue this option. (If your work is more suitable for medRxiv, you will need to post the preprint yourself, as the mechanisms for us to do so are still in development.)

Summary:

Kazuki and colleagues report their generation of a new transchromosomal mouse model of Down syndrome with a freely segregating copy of the long arm of human chromosome 21 (HSA21q). This model (MAC21) recapitulates the change seen in Down syndrome and will allow analyses without viability and fertility issues. To begin to characterize this model the authors have analyzed gene expression and found that unlike the current Tc1 model, mice are fully trisomic for HSA21q. Like Tc1 mice, MAC21 mice have growth retardation, abnormalities in heart and facial development, and hematological abnormalities. However in contrast to current Down Syndrome models, MRI studies of the brains of MAC21 and control mice indicate that the MAC21 brain (except the cerebellum) is larger than controls, a finding at odds with other mouse models and with the microcephaly reported in the human brain. The authors also assayed several well-studied cellular and behavioral phenotypes in people with DS and in other mouse models. In many cases, their results are similar to those published previously in human and mouse model studies, but there are findings that are discordant with current mouse models. These findings may indicate the novelty and importance of this new model and should be commented on in the text.

Essential revisions:

1) Previous studies on human and mouse samples indicate that trisomy 21 (or trisomy 16 in the mouse models) is correlated with destabilization of genes expressed from other chromosomes in the genome. It will be important to know whether the increased complement of HSA21 genes in the MAC21 mouse also affects up- or down-regulation of genes throughout the genome. I assume these data are in hand based on the results for Figure 2.

2) Although the sequence of Hsa21 is made available, it would be important to describe the origin of the Hsa21 and to introduce the main allelic content of the chromosome in the paper. Only one allelic variant is described for SON.

3) Almost no details are described for the Cre-dependent reciprocal translocation in the Results or the Materials and methods sections. More information are needed including the frequency of the event.

4) The behavioural phenotypes have been scored with males and females but almost no information about the order of the test, the age of the mouse, the time between test are provided (only for nesting). The authors should follow the ARRIVE guidelines https://www.nc3rs.org.uk/arrive-guidelines to make their new model accessible and their analyse more reproducible.

5) In all behavioral assays both sexes should be reported separately. In the Figure 6 and some panel of Figure 7, the data seems to be combined for both sexes. It would be better to give separated figures per sex. Thus, Figure 6D should be shown for male and female separately and you can remove Figure 6E that is detailed in Figure 6F.

6) For the MWM, data should be presented by sexes as done in Figure 7B. Indeed, only MAC21 males were delayed in finding the platform. To avoid confusion Figure 7A, C, E, I should be removed as they reported both sexes together and a sex difference is found on Figure 7B. As reported both MAC21 carrier and controls were able to remember where the platform was although with a slight difference, especially after 4 days of trials. The statistical analysis should be clarified if they were done at the last day or during the probe test in the Figure 7D, and F. thus the conclusion of the paragraph should be moderated: The spatial memory is still operational in the TcMac21. Thus, "Disruption" may be a too strong word here.

7) In the Table 2, that summarize many DS models, to be fair, the authors should clearly remind the genetic differences observed in their model where 14 genes are not trisomic (Figure 1F? deletion in the MAC21).

8) The authors should explain why in the RRWM only females were tested? The statistical analysis in the Figure 7J should be made more explicit. It is difficult to understand what has been tested with the ANOVA compared to the other panels.

9) Please explain how the following statement is supported: "In reversal 1, TcMAC21 performed relatively worse than Eu in acquisition trials (p=0.15, Figure. 7G). if p=0.15 both genotypes performed similarly, did not they?

---

## [Author Response]

Essential revisions:1) Previous studies on human and mouse samples indicate that trisomy 21 (or trisomy 16 in the mouse models) is correlated with destabilization of genes expressed from other chromosomes in the genome. It will be important to know whether the increased complement of HSA21 genes in the MAC21 mouse also affects up- or down-regulation of genes throughout the genome. I assume these data are in hand based on the results for Figure 2.

We analyzed the effects of HSA21 on all mouse gene expression and generated Figure 2—source data 2, and then in the Results section, we added that “To test effects of HSA21q on gene expression of other mouse chromosomes, we analyzed gene expression changes among 13976 mouse genes (both non-coding and coding) whose FPKM ≥1 in Eu (Figure 2—source data 2). We found 712 of these genes in TcMAC21 were down-regulated (TcMAC21/Eu<0.8) and 1191 genes were up-regulated (TcMAC21/Eu>1.2), indicating that as in other mouse models and people with trisomy 21 steady state RNA levels are perturbed throughout the genome.”

2) Although the sequence of Hsa21 is made available, it would be important to describe the origin of the Hsa21 and to introduce the main allelic content of the chromosome in the paper. Only one allelic variant is described for SON.

To the degree that it is known, the origin of Hsa21 is described in the manuscript (Tomizuka et al., 1997 and Inoue et al., 2001). Note that while the origin of the chromosome is certainly of interest and is addressed to the extent to which it is known, the sequence of this chromosome is not expected to be identical after 25 years of cell culture in various artificial contexts. We believe that “Figure 1—source data 1D” from the original submission, which contains detailed information of all 91 single nucleotide variations (SNVs) in HSA21q PCGs detected by deep coverage WGS, provides the most relevant reference for those interested in specific gene effects in TcMAC21. We found that 90 of the 91 were previously reported as native alternative alleles, making it unlikely that wholesale mutation in culture has modified the chromosome (Figure 1—source data 1D). As we present in the manuscript, the only SNV not previously identified is in the SON gene and that the variant is predicted to be a tolerated mutation.

3) Almost no details are described for the Cre-dependent reciprocal translocation in the Results or the Materials and methods sections. More information are needed including the frequency of the event.

In the Results section, we added that “The recombinant clones were selected using HAT, and 16 out of the 18 drug-resistant clones were PCR-positive with Cre-loxP recombination-specific primers. Two lines out of the examined 8 clones were confirmed by FISH to contain HSA21q-MAC”.

4) The behavioural phenotypes have been scored with males and females but almost no information about the order of the test, the age of the mouse, the time between test are provided (only for nesting). The authors should follow the ARRIVE guidelines https://www.nc3rs.org.uk/arrive-guidelines to make their new model accessible and their analyse more reproducible.

In the Materials and methods section, “3-month-old mice (females: Eu=13, TcMAC21 =11; males: Eu =11, TcMAC21 =12) were used for behavioral tests in the sequence, open field, visual discrimination water maze test, MWM, and RRWM.” Thank you for catching our unintentional deletion of a sentence containing the age information during editing. This information is restored in the Materials and methods and added to figure legends. To more prominently present this information, we also added “Figure 6—figure supplement 1” to show the experimental design for behavioral tests and electrophysiology.

We followed the ARRIVE guidance, and generated a new consolidated table of demographic animal information, including genetic background, age, gender, and animal number for control and TcMAC21, for each experiment of each figure (Table 3), to clarify interpretation of procedures involving the new model. In each figure and figure legend, we further reemphasized these animal information and provided statistical methods and results. The details of animal generation and experimental procedures were described in the Materials and methods and Results sections.

5) In all behavioral assays both sexes should be reported separately. In the Figure 6 and some panel of Figure 7, the data seems to be combined for both sexes. It would be better to give separated figures per sex. Thus, Figure 6D should be shown for male and female separately and you can remove Figure 6E that is detailed in Figure 6F.

Agree, Figure 6 has been modified accordingly.

6) For the MWM, data should be presented by sexes as done in Figure 7B. Indeed, only MAC21 males were delayed in finding the platform. To avoid confusion Figure 7A, C, E, I should be removed as they reported both sexes together and a sex difference is found on Figure 7B. As reported both MAC21 carrier and controls were able to remember where the platform was although with a slight difference, especially after 4 days of trials. The statistical analysis should be clarified if they were done at the last day or during the probe test in the Figure 7D, and F. thus the conclusion of the paragraph should be moderated: The spatial memory is still operational in the TcMac21. Thus, "Disruption" may be a too strong word here.

Agree. Figure 7A, C and E have been removed and replaced to generate a new Figure 7 with updated Results section (subsection “TcMAC21 mice have significant learning and memory deficits”).

We would like to clarify that the statistical analysis of initial Figure 7A, C, E took both HSA21 and gender effects into account as we used repeated measures ANOVA and Fisher’s least significant difference (LSD) post hoc test to analyze effects of HSA21, gender, their interaction and provided statistical tables (Figure 7—source data 1) showing their effects on MWM outcomes: Figure 7A “acquisition trials” (HSA21: p=0.015, gender: p=0.53, the interaction: p=0.53); Figure 7C “short delay probe trials” (HSA21: p<0.002, gender: p=0.55, and the interaction: p=0.59); Figure 7E “long delay probe trials” (HSA21: p<0.0001, gender: p=0.01, the interaction: p=0.38). Gender effects are quite small compared with HSA21 effects. The advantage of this analysis over the separate analysis by gender is more statistical power because of larger sample size. As MWM is the most important spatial memory test to analyze this new model, we move (Figure 7A, C and E) into Figure 7—figure supplement 2 that helps readers to determine the sample size when they use this model.

We remove “disruption” in favor of the terms “deficit” or “impairment” as suggested.

7) In the Table 2, that summarize many DS models, to be fair, the authors should clearly remind the genetic differences observed in their model where 14 genes are not trisomic (Figure 1F? deletion in the MAC21).

This information is throughout the text and appears (clearly, we believe) in Table 2. The row “Trisomic HSA21q PCG/orthologs (% of 213 HSA21q)” shows that TcMAC21 contained 199 (=213-14) HSA21q PCG or 93% of 213 HSA21q PCG in Table 2.

8) The authors should explain why in the RRWM only females were tested? The statistical analysis in the Figure 7J should be made more explicit. It is difficult to understand what has been tested with the ANOVA compared to the other panels.

The choice of females for RRWM was purely pragmatic. RRWM has not been reported widely in DS models, but we were aware that “previous study showed that female but not male APP/PS1 mice showed deficits in reversal water maze (Gallagher, Minogue and Lynch, 2013)”. The sentence inside the quotation marks has been added in the Results section.

To better explain original Figure 7J (Quantitative analysis of forgetting old platform SE position), we updated RRWM result section (subsection “TcMAC21 mice have significant learning and memory deficits”).

9) Please explain how the following statement is supported: "In reversal 1, TcMAC21 performed relatively worse than Eu in acquisition trials (p=0.15, Figure 7G). if p=0.15 both genotypes performed similarly, did not they?

Agree, we revised the descriptions in RRWM.